# Structure of PDE3A-SLFN12 complex reveals requirements for activation of SLFN12 RNase

Colin W. Garvie[1,12], Xiaoyun Wu[2,12], Malvina Papanastasiou [3], Sooncheol Lee[2], James Fuller[4], Gavin R. Schnitzler[2], Steven W. Horner [1], Andrew Baker[2], Terry Zhang[5], James P. Mullahoo [3], Lindsay Westlake[2], Stephanie H. Hoyt [2], Marcus Toetzl[2], Matthew J. Ranaghan [1], Luc de Waal[2], Joseph McGaunn [2], Bethany Kaplan[2], Federica Piccioni[6,10], Xiaoping Yang[6], Martin Lange[7,11], Adrian Tersteegen[8], Donald Raymond[1], Timothy A. Lewis[1], Steven A. Carr [3], Andrew D. Cherniack [2,9], Christopher T. Lemke[1], Matthew Meyerson [2,9] & Heidi Greulich [2,9✉]

DNMDP and related compounds, or velcrins, induce complex formation between the phosphodiesterase PDE3A and the SLFN12 protein, leading to a cytotoxic response in cancer cells that express elevated levels of both proteins. The mechanisms by which velcrins induce complex formation, and how the PDE3A-SLFN12 complex causes cancer cell death, are not fully understood. Here, we show that PDE3A and SLFN12 form a heterotetramer stabilized by binding of DNMDP. Interactions between the C-terminal alpha helix of SLFN12 and residues near the active site of PDE3A are required for complex formation, and are further stabilized by interactions between SLFN12 and DNMDP. Moreover, we demonstrate that SLFN12 is an RNase, that PDE3A binding increases SLFN12 RNase activity, and that SLFN12 RNase activity is required for DNMDP response. This new mechanistic understanding will facilitate development of velcrin compounds into new cancer therapies.

[1] Center for the Development of Therapeutics, Broad Institute of MIT and Harvard, Cambridge, MA, USA. [2] Cancer Program, Broad Institute of MIT and Harvard, Cambridge, MA, USA. [3] Proteomics Platform, Broad Institute of MIT and Harvard, Cambridge, MA, USA. [4] Helix Biostructures, Indianapolis, IN, USA. [5] Thermo Fisher, San Jose, CA, USA. [6] Genetic Perturbation Platform, Broad Institute of MIT and Harvard, Cambridge, MA, USA. [7] Research and Development, Pharmaceuticals, Bayer AG, Berlin, Germany. [8] Research and Development, Pharmaceuticals, Bayer AG, Wuppertal, Germany. [9] Department of Medical Oncology, Dana-Farber Cancer Institute, Boston, MA, USA. [10] Present address: Merck Research Laboratories, Boston, MA, USA. [11] Present address: NUVISAN ICB GmbH, Berlin, Germany. [12] These authors contributed equally: Colin W. Garvie, Xiaoyun Wu. ✉email: heidig@broadinstitute.org

A class of small molecules has recently been described that causes selective cancer cell killing by inducing complex formation between two cellular proteins, PDE3A and SLFN12[1–3]. These small molecules are exemplified by the prototypical PDE3A-SLFN12 complex inducer, DNMDP[1], although other classes of PDE3A-SLFN12 complex inducers have subsequently been described with similar, albeit weaker, activity[4–7]. Cancer cells expressing elevated levels of both PDE3A and SLFN12 are typically sensitive to killing by DNMDP and other PDE3A-SLFN12 complex inducers.

PDE3A is a well-characterized cyclic nucleotide phosphodiesterase (PDE) that hydrolyzes cAMP, cGMP, and cUMP[8,9]. DNMDP is a PDE3A inhibitor and the catalytic domain of PDE3A is sufficient to support response to DNMDP[1,3]. However, the cancer cell killing activity of DNMDP does not correlate with inhibition of PDE3A enzymatic activity, in that other potent and selective PDE3 inhibitors such as trequinsin[10] do not kill cancer cells, and knockout of PDE3A from sensitive cell lines abolishes DNMDP sensitivity[1,3]. DNMDP instead has a gain- or change-of-function effect on PDE3A, involving induction of complex formation with SLFN12.

Unlike PDE3A, little is known about the normal physiological function of SLFN12 beyond its expression in T cells and association of ectopic expression with differentiation and/or quiescence[11–14]. Most human Schlafen genes, including *SLFN5*, *SLFN11*, *SLFN13*, and *SLFN14*, are significantly longer than *SLFN12* and encode a C-terminal helicase domain and a nuclear localization signal, which *SLFN12* and the related *SLFN12L* do not encode[15]. The six human SLFN genes have a semi-conserved SWADL motif in common[16], as well as a divergent AAA ATPase domain that may function as an RNA-binding domain[17].

We took multiple orthogonal approaches to determine how DNMDP-like molecules induce PDE3A-SLFN12 complex formation. We first solved the crystal structure of the PDE3A catalytic domain bound to several ligands, including DNMDP and trequinsin. This structure was used to map intramolecular and intermolecular changes in PDE3A upon complex formation with SLFN12 using hydrogen–deuterium exchange mass spectrometry (HDX-MS). The full cryo-electron microscopy (Cryo-EM) structure of the PDE3A-SLFN12 complex with bound DNMDP revealed the molecular details of complex formation and the role of DNMDP in stabilizing the complex. Deep mutational scanning (DMS) of PDE3A identified amino acids required for DNMDP response, including residues involved in compound binding, PDE3A homodimerization, and SLFN12 binding, substantiating findings from the structural studies. Finally, we show that, like other SLFN family members[18–20], SLFN12 is an RNase, that RNase activity is increased upon PDE3A binding, and that SLFN12 RNase activity is required for DNMDP response.

## Results

### DNMDP induces complex formation between purified PDE3A and SLFN12.

Previous work showed that DNMDP induces complex formation between PDE3A and SLFN12 in cells[1]. To determine whether this complex can be recapitulated with isolated proteins, we expressed and purified the catalytic domain of PDE3A (PDE3A$^{CAT}$), which comprises residues 640–1141, and full-length SLFN12 and analyzed their ability to interact in the absence and presence of DNMDP. We limited our analysis to the catalytic domain of PDE3A because our previous experiments indicated that the N-terminal portion of PDE3A, containing several membrane association domains[21], was not required for DNMDP sensitivity in cells[3]. Analysis of recombinant PDE3A$^{CAT}$ by analytical size exclusion chromatography (SEC) revealed a single species at an elution volume earlier than expected for a

monomer of theoretical mass 57.3 kDa (Fig. 1a). The solution mass of PDE3A$^{CAT}$ was determined to be 118 kDa using multi-angle light scattering (MALS) as the protein eluted from a size exclusion column (SEC-MALS), showing that the catalytic domain existed as a dimer (Supplementary Fig. 1). Dimerization of PDE3A$^{CAT}$ was supported by sedimentation equilibrium analytical ultracentrifugation (SE-AUC) data, which could be fit to a monomer–dimer equilibrium with a $K_d$ of 40 nM (Fig. 1b). Considering that PDE3A contains additional regions N-terminal to the catalytic domain that are proposed to promote oligomerization[21], it is likely that the $K_d$ for dimerization of full-length PDE3A protein is even lower than observed here. Purified SLFN12 showed concentration-dependent aggregation at low NaCl concentrations and was stored and analyzed at 500 mM NaCl, except where noted. Analysis by SEC revealed that SLFN12 eluted significantly later than would be expected for a protein of theoretical mass 67.3 kDa, suggesting a non-specific interaction with the column resin (Fig. 1a). Measurement of the solution mass of SLFN12 by SEC-MALS gave a mass of 126.9 kDa, showing that it also existed as a dimer at micromolar protein concentrations (Supplementary Fig. 1). SE-AUC analysis of SLFN12, however, showed that the dimer was significantly less stable than observed for PDE3A$^{CAT}$, with the data fitting to a monomer–dimer $K_d$ of 870 nM (Fig. 1c).

To investigate the interaction between PDE3A and SLFN12, a pulldown assay was performed with a version of SLFN12 with maltose-binding protein (MBP) fused to its N-terminus. MBP-SLFN12 was immobilized on amylose resin and incubated with PDE3A$^{CAT}$ in the absence and presence of DNMDP at 150 mM NaCl (Fig. 1d). Under these conditions, PDE3A$^{CAT}$ was able to bind MBP-SLFN12 in the absence of compound, and the quantity bound was only slightly increased in the presence of DNMDP. In contrast, trequinsin, a potent and selective PDE3A/B inhibitor (Supplementary Table 1) with no cancer cell killing activity[1], inhibited binding of PDE3A to MBP-SLFN12. Curiously, when the experiment was repeated at 500 mM NaCl, less PDE3A$^{CAT}$ interacted with MBP-SLFN12 except in the presence of DNMDP, which dramatically increased complex formation (Fig. 1d).

SEC was used to further investigate complex formation between PDE3A and SLFN12 and the effect of DNMDP. At physiological salt concentration (150 mM NaCl), the two proteins coeluted at an earlier volume than either protein alone, indicative of stable complex formation (Fig. 1a). Addition of DNMDP had only a minor effect on complex formation under these conditions. However, at 500 mM NaCl, the earlier eluting complex was stably formed only in the presence of DNMDP (Fig. 1e). In contrast, the chromatogram of the PDE3A-SLFN12 run in the absence of DNMDP was consistent with partial dissociation into constituent PDE3A and SLFN12 proteins. When measured by SEC-MALS, the mass of the PDE3A$^{CAT}$ + SLFN12 complex at 150 mM NaCl was determined to be 246.9 kDa, in close agreement with a dimer of PDE3A$^{CAT}$ interacting with a dimer of SLFN12 (Supplementary Fig. 1).

To quantify the effect of DNMDP on complex formation, we immobilized biotinylated Avi-tagged PDE3A$^{CAT}$ on a streptavidin biosensor and measured its interaction with SLFN12 using biolayer interferometry (BLI; Fig. 1f, g). These experiments could only be performed at 500 mM NaCl as SLFN12 showed non-specific binding to the sensors at 150 mM. We observed clear association between SLFN12 and the immobilized PDE3A. Addition of DNMDP increased the rate of association and reduced the rate of dissociation of SLFN12 with PDE3A. The very slow dissociation rate suggests that the complex is highly stable over time. Trequinsin, as expected, inhibited binding. The calculated steady state $K_d$ for complex formation at 500 mM NaCl in the absence and presence of DNMDP was 320 and 65 nM, respectively (Fig. 1g).

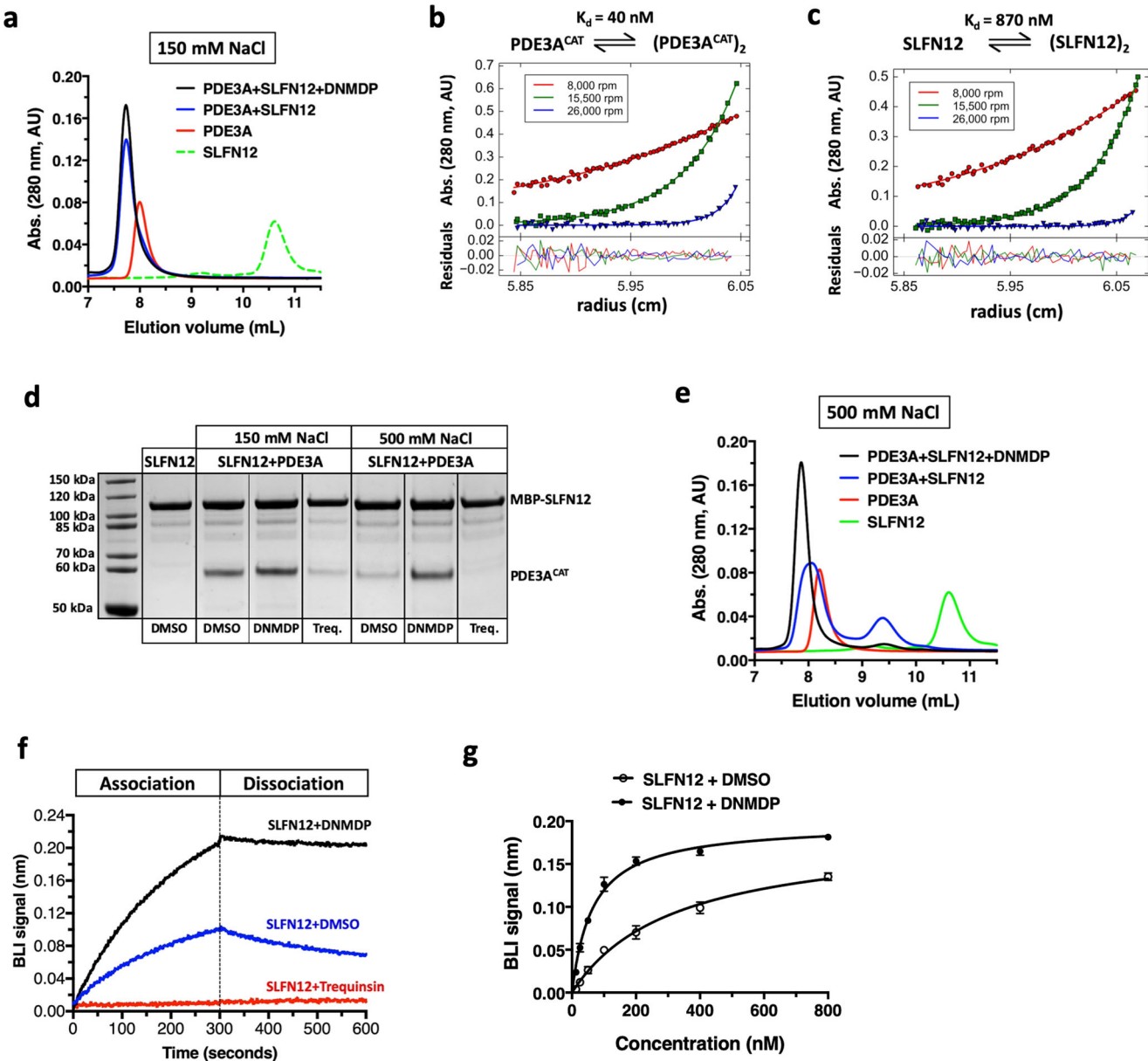

**Fig. 1 DNMDP induces complex formation between purified PDE3A and SLFN12. a** SEC analysis of complex formation performed at 150 mM NaCl. Traces are shown for 10 μM PDE3A$^{CAT}$ (red), SLFN12 (green), and PDE3A$^{CAT}$ + SLFN12 in the absence (blue) and presence of 100 μM DNMDP (black). The trace for SLFN12 had to be collected at 500 mM NaCl and is therefore shown as a dashed line. **b** SE-AUC analysis of PDE3A$^{CAT}$ and **c** SLFN12. The solid lines represent the best fit to a monomer–dimer model with the $K_d$ indicated. **d** Amylose resin pulldown analysis of complex formation. SDS-PAGE gel of protein(s) eluted from amylose resin when 2 μM of His$_6$-MBP-SLFN12 was incubated alone or with 2 μM PDE3A$^{CAT}$ ± 10 μM DNMDP or trequinsin at 150 or 500 mM NaCl. **e** SEC analysis of complex formation performed at 500 mM NaCl. **f** Effect of DNMDP and trequinsin on complex formation by BLI. Binding of 100 nM SLFN12 to immobilized PDE3A$^{CAT}$ in the presence of 500 mM NaCl and DMSO (blue), 10 μM DNMDP (black), or 10 μM trequinsin (red). **g** Quantitative analysis of the effect of DNMDP on complex formation using BLI. Binding to SLFN12 performed in the absence (open circles) and presence (filled circles) of 100 μM DNMDP at 500 mM NaCl. Data were analyzed from four independent experiments and represented as mean values ± s.e.m. Abs. absorbance, Treq. trequinsin.

Taken together, these data demonstrate that purified PDE3A$^{CAT}$ and SLFN12 are sufficient to form a stable complex comprised of a dimer of PDE3A and a dimer of SLFN12 and that DNMDP significantly stabilizes the complex. The co-chaperone, aryl hydrocarbon receptor interacting protein (AIP), is required for PDE3A-SLFN12 complex formation in cells[3]. However, AIP is evidently not required for complex formation in vitro. The influence of salt concentration on complex formation suggested that electrostatic interactions play an important role in promoting complex formation.

**DNMDP and trequinsin have only a limited effect on the structure of PDE3A$^{CAT}$.** DNMDP was previously shown to be an inhibitor of PDE3A PDE activity, implying that DNMDP binds directly to PDE3A[1]. Indeed, the melting temperature ($T_m$) of PDE3A$^{CAT}$ increased by +3 °C in the presence of DNMDP (Supplementary Fig. 1), indicating that DNMDP binds and stabilizes the structure of PDE3A$^{CAT}$. In contrast, no change in $T_m$ was observed upon incubating SLFN12 with DNMDP (Supplementary Fig. 1). This suggests that SLFN12 does not bind to DNMDP in the absence of PDE3A, although we cannot discount

the possibility that its binding does not sufficiently impact the stability of the structure to cause a change in the melting temperature of SLFN12.

To gain insight into how DNMDP induces complex formation of PDE3A with SLFN12 and how trequinsin inhibits this interaction, we solved the high-resolution crystal structure of a modified form of the catalytic domain of PDE3A (PDE3A$^{CAT-Xtl}$) in the absence or presence of DNMDP or trequinsin (Fig. 2a and Supplementary Table 2). PDE3A$^{CAT-Xtl}$ is comprised of residues 669–1095 with two internal loops between residues 780–800 and 1029–1067 replaced with shorter linkers to aid in crystallization and improve diffraction quality of the crystals (Fig. 2a). We also obtained the structure of AMP bound to PDE3A by soaking the apo-crystals with cAMP, demonstrating that the crystalline form of PDE3A$^{CAT-Xtl}$ was catalytically active, similar to PDE4D2 crystallized with cAMP[22]. The catalytic domain of PDE3A crystallized as a dimer with two dimers in the asymmetric unit (Fig. 2a). The structure is very similar to the structure of the catalytic domain of PDE3B[23], with which it shares 69% sequence identity, with a root mean square deviation (RMSD) of 0.53 Å for the main chain atoms. The PDE3A crystal structures are essentially identical, with only small movements of the protein backbone at the active site to accommodate the binding of AMP, DNMDP, and trequinsin (Supplementary Fig. 1).

In the AMP-bound structure, the AMP is stretched across the catalytic site, with the phosphate group coordinating directly with the two metal ions at one end and the purine moiety making hydrogen bonds with Q1001 at the other end (Fig. 2b). The adenine moiety is further stabilized by hydrophobic contacts with I968 and π–π stacking with F1004. In the DNMDP-bound structure, DNMDP binds in an extended conformation with the dihydropyrazidinone ring buried deep in the active site pocket (Fig. 2c). This orientation is stabilized by hydrogen bonds with H961 and Q1001 and a series of hydrophobic interactions along the length of DNMDP. F972 and L910 also pack against each of the ethyls of the diethylamino group of DNMDP, which are positioned at the entrance of the active site pocket. Trequinsin binds in a similar location as DNMDP, with the aromatic ring at the di-methoxy end being roughly superimposable on the phenyl ring of DNMDP; however, it is not as deeply embedded into the active site and is stabilized only by non-polar interactions (Fig. 2d). The trimethylphenyl group of trequinsin, which is rotated 90° relative to the plane of the rest of the molecule, additionally packs into a hydrophobic pocket created by the Cβ of S1003 and several hydrophobic side chains at the entrance of the catalytic site.

It is clear from the crystal structures that DNMDP and trequinsin inhibit PDE3A catalytic activity by sterically restricting entry of cAMP and preventing key stabilizing contacts. However, they do not reveal how DNMDP promotes SLFN12 binding, as DNMDP does not cause any obvious structural changes in PDE3A. We hypothesized that the interaction with SLFN12 is stabilized by contacts with the exposed diethylamino group of DNMDP and inhibited by the trimethylphenyl group of trequinsin.

**HDX-MS identified three regions of PDE3A$^{CAT}$ with decreased solvent exposure following SLFN12 binding.** To gain further insight into PDE3A-SLFN12 complex formation, we utilized HDX-MS to identify regions of PDE3A that were affected upon binding to SLFN12. HDX-MS provides an excellent way to probe protein–protein interactions in solution by comparing the relative deuterium uptake between amide hydrogens of the protein backbone in the protein alone and bound to a partner[24]. Experiments were performed with PDE3A$^{CAT}$ bound to the DNMDP analog BRD9500[2] in the absence and presence of

SLFN12, using a high salt concentration (500 mM) to ensure a structurally uniform SLFN12 population.

Analysis of the uptake of deuterium into PDE3A$^{CAT}$ in the presence of BRD9500 revealed many slowly exchanging regions, which corresponded to well-folded regions of the protein observed in the crystal structure (Supplementary Fig. 2 and Supplementary Data 1). Extensive deuteration at the earliest time point was observed in both the N- and C-terminals of the protein and in the loop regions between 778–793 and 1028–1068, which indicates dynamic regions.

The deuterium uptake profiles for PDE3A$^{CAT}$ bound to BRD9500 were then compared to data collected in the presence of SLFN12, focusing on the regions that show either an increase or decrease in deuterium exchange. In the presence of SLFN12, three distinct regions of PDE3A$^{CAT}$ exhibited a decrease in exchange of deuterium: region 1 covering residues 849–867, region 2 covering residues 902–940, and region 3 covering residues 983–1001 (Fig. 2e, Supplementary Fig. 2, and Supplementary Data 1). A decrease in deuterium exchange is often associated with the region becoming shielded from the solvent, potentially as a result of interacting with a partner protein. This is likely the case with solvent-exposed regions 2 and 3. Intriguingly, they are also at or near the DNMDP-binding site, supporting the hypothesis that SLFN12 binds close to this region. Region 1, which encompasses peptides at the PDE3A$^{CAT}$ dimerization interface, exhibited the greatest decrease in deuterium exchange. This suggests that the binding of SLFN12 stabilizes the PDE3A$^{CAT}$ homodimer further, presumably via the reduction of structural fluctuations in the dimer interface, not captured in the crystal structure. Under the experimental conditions employed, we did not detect any long-distance allosteric changes in PDE3A upon interaction with SLFN12, presumably with regions 2 and 3, although the deuterium changes detected in the PDE3A dimer interface may be considered such.

**Cryo-EM solution of the PDE3A-SLFN12 complex structure.** To directly address how DNMDP promotes PDE3A-SLFN12 complex formation, we used Cryo-EM to solve the structure of the PDE3A$^{CAT}$-DNMDP complex bound to full-length SLFN12 (Supplementary Fig. 3 and Supplementary Table 3). The complex was found to possess twofold symmetry, which was applied throughout refinement, and consisted of two flexibly connected bodies (Supplementary Movie 1). The bodies could be readily identified by fitting with the PDE3A$^{CAT-Xtl}$ dimer crystal structure and two monomers of the N-terminal domain (NTD) of rat SLFN13, which shares 38% sequence identity with the related region of SLFN12[20]. The dynamic motion of the two bodies necessitated refinement of each protein dimer separately utilizing the multi-body refinement approach in RELION[25], with the final resolution for the maps for SLFN12 and PDE3A$^{CAT}$ being 2.76 and 2.97 Å, respectively (Supplementary Figs. 3 and 4). Residues 669–1100 for each monomer of PDE3A$^{CAT}$ could be modeled except for the loop regions between 779–799 and 1029–1068, which likely adopted multiple conformations. These loop regions are equivalent to the ones that were replaced by short linkers in the crystal structure. Residues 1–560 of each monomer of SLFN12 could be modeled, with the first 346 residues being guided by the structure of rat SLFN13 and residues 387–560 built de novo. The regions between 346 and 386 and beyond residue 561 were not visible in the maps, suggesting that they adopted multiple conformations. There was also clear density for two DNMDP molecules, one in the catalytic site of each PDE3A monomer (Supplementary Fig. 4).

In agreement with the biophysical studies, the Cryo-EM structure shows a dimer of PDE3A$^{CAT}$-DNMDP interacting with

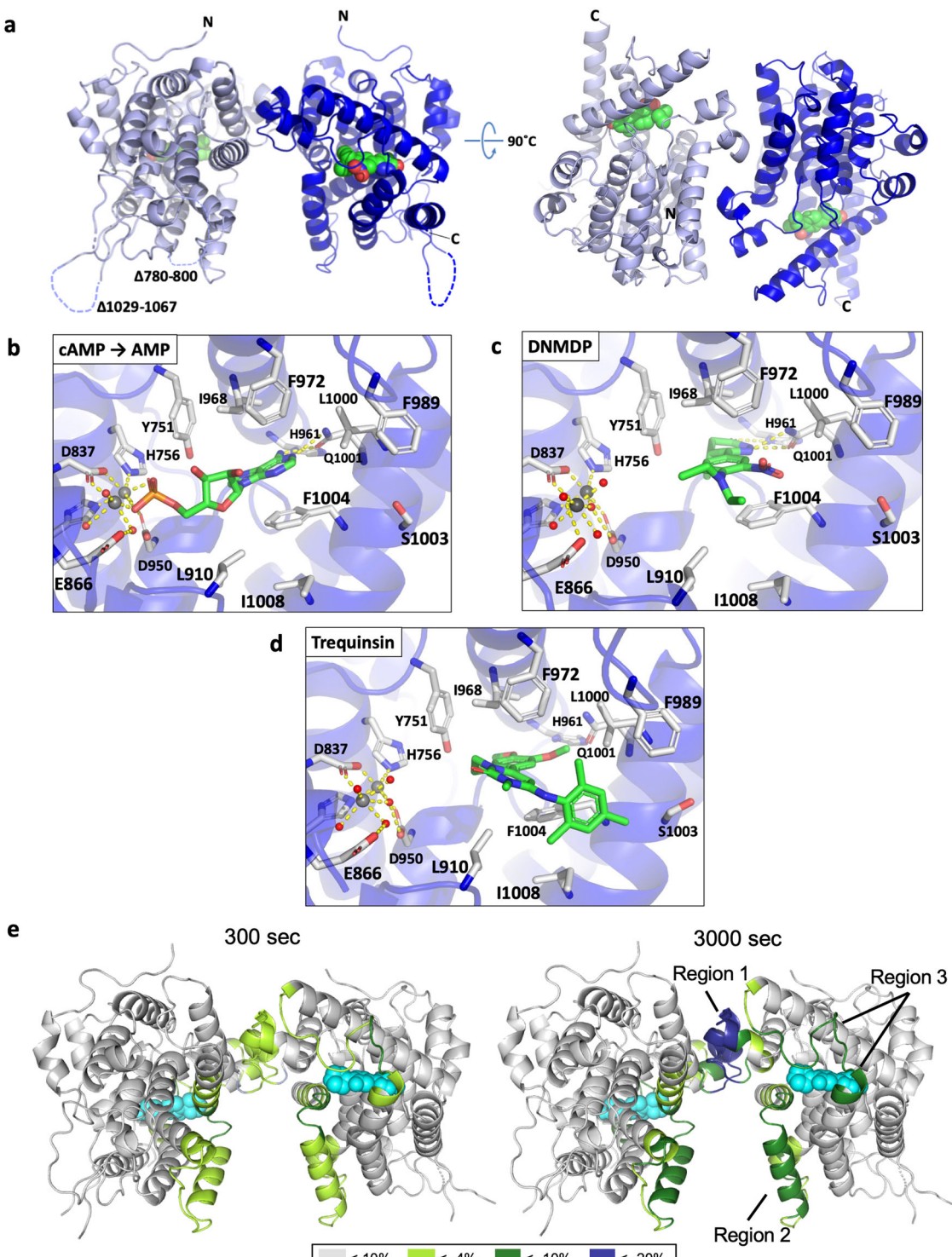

**Fig. 2 DNMDP has only a limited effect on the structure of PDE3A$^{CAT}$ but HDX-MS reveals regions of PDE3A$^{CAT}$ with decreased solvent exposure following SLFN12 binding. a** Crystal structure of PDE3A$^{CAT-Xtl}$ bound to DNMDP. Each monomer is colored dark and light blue. DNMDP is shown in space-filling for clarity with the carbon, oxygen, and nitrogen atoms colored green, red, and blue, respectively. The two loop regions replaced by linkers are indicated by dashed lines. **b**-**d** Catalytic sites from the PDE3A$^{CAT-Xtl}$ crystal structures. **b** PDE3A$^{CAT-Xtl}$-AMP; **c** PDE3A$^{CAT-Xtl}$-DNMDP; and **d** PDE3A$^{CAT-Xtl}$-trequinsin are shown. The side chains are shown in a licorice format, with the carbon atoms colored white. The AMP, DNMDP, and trequinsin are shown in a similar format except the carbons that are colored green for visualization purposes. The phosphorus of AMP is shown in orange. The two metal ions and the coordinating water molecules are represented as gray and red spheres, respectively. Hydrogen bonds and metal–ligand interactions are shown as dashed yellow lines. **e** Analysis of the PDE3A$^{CAT}$ and SLFN12 interface by HDX-MS. D-uptake differences are mapped with green and blue onto the PDE3A$^{CAT-Xtl}$ crystal structure for 300 and 3000 s. BRD9500 is depicted with cyan spheres.

a dimer of SLFN12 (Fig. 3a). The structure of PDE3A$^{CAT}$ is essentially the same as PDE3A$^{CAT-Xtl}$ (RMSD for backbone atoms of 0.38 Å), with only an additional two turns of the C-terminal helices for each monomer modeled (Supplementary Fig. 5). There was no density evident for the loop regions between residues 779–799 and 1029–1068, suggesting that they adopt multiple conformations and are not involved in contacting SLFN12. In the HDX-MS studies, these regions showed a high deuterium content at the earliest time point in the absence of SLFN12, indicating that they are dynamic (Supplementary Fig. 2). The deuterium uptake did not change in the presence of SLFN12, supporting the observation that they are not affected by complex formation (Supplementary Fig. 2).

The structure of the SLFN12 monomers can be divided into an NTD and C-terminal domain (CTD), which comprise residues 1–345 and 387–560, respectively (Fig. 3b). The SLFN12-NTD can be further sub-divided into an N-lobe, C-lobe, and two bridging domains (BDs; Fig. 3b and Supplementary Fig. 5). The overall structure of the SLFN12-NTD is very similar to the rat SLFN13-NTD, with an RMSD of main chain atoms of 0.55 and 0.75 Å for the individual N-lobes and C-lobes, respectively. The SLFN12-CTD is comprised of a core domain between residues 387–541 and the PDE3A interacting region (PIR), between residues 551 and 560 (Fig. 3b and Supplementary Fig. 5). The linker between these two regions gives rise to the flexibility observed between the PDE3A and SLFN12 dimers. This linker is stabilized by hydrophobic interactions between F548, other residues in the linker, and the CTD core region (Fig. 3b and Supplementary Fig. 5). While the SLFN12-NTD and -CTD are connected by a flexible linker, they are stabilized relative to each other by a series of hydrophobic and salt bridge interactions (Supplementary Fig. 5). The two SLFN12 monomers interact over the length of the NTD and a small portion of the CTD (Fig. 3b). The interactions are centered around four residues from each monomer: T71, F89, I131, and I517. The side chains of these four amino acids are inserted into hydrophobic pockets formed by residues from the opposite monomer (Fig. 3c).

**The C-terminal helix of SLFN12 is primarily responsible for complex formation.** The majority of the interactions between PDE3A$^{CAT}$ and SLFN12 occurs via the PIR sequence 551-AENLYQIIGI-560 at the C-terminal end of each SLFN12 monomer, which extends out from the main body of SLFN12 (Figs. 3 and 4a). Almost all of the amino acids in this sequence interact directly with residues around the entrance of the catalytic site of PDE3A through hydrophobic interactions (Fig. 4b). This includes a close packing interaction between one of the ethyls of the diethylamino group of DNMDP and I557 of SLFN12, which provides a direct explanation for the increase in stability of the PDE3A-SLFN12 complex in the presence of DNMDP (Fig. 4b). It is interesting to note that several PDE3A residues are involved in contacting both DNMDP and SLFN12 (Supplementary Table 4). It is possible that the binding of DNMDP may help stabilize the position of these residues to optimize interaction with SLFN12. Modeling the position of trequinsin into the Cryo-EM structure, based on its location in the crystal structure, clearly shows that it would sterically clash with the L554, I557, and I558 side chains of SLFN12, making the interaction between PDE3A and SLFN12 highly unstable (Fig. 4c). Outside of the C-terminal region of SLFN12, the only other point of contact between the two proteins occurs through a short loop from each PDE3A monomer that contains two acidic residues, D926 and D927, which point toward a highly positively charged surface of SLFN12 (Fig. 4a, d). D927 from PDE3A forms a direct salt bridge with K150 from SLFN12 from each monomer at this interface (Fig. 4a). This may in part

explain why binding between PDE3A and SLFN12 was destabilized at higher salt concentrations.

The importance of the SLFN12 PIR to PDE3A-SLFN12 complex formation was supported by deletion studies. Whereas a 10-amino acid C-terminal truncation did not impair DNMDP response when transduced into HeLa-Res cells lacking endogenous SLFN12 expression, a 30-amino acid truncation removing the PIR domain, but with similar levels of expression, completely abolished DNMDP response (Fig. 5a–c). These results demonstrate a requirement of the SLFN12 PIR for response to DNMDP.

**Deep Mutational Scanning of PDE3A identifies DNMDP resistance mutations.** To further investigate the structural relationship between PDE3A, DNMDP, and SLFN12, we used DMS to identify residues of PDE3A that impact DNMDP sensitivity. Because we previously showed that the isolated catalytic domain of PDE3A was sufficient to confer DNMDP sensitivity in cells expressing SLFN12 but lacking endogenous PDE3A[3], we limited our mutational analysis to the PDE3A catalytic domain. We designed a library of PDE3A alleles in which the sequence encoding amino acids 668–1141, including the catalytic domain, was substituted with a codon for every other possible amino acid or a stop codon in the context of the full-length cDNA (Supplementary Data 2). This library was expressed in PDE3A-knockout GB1 glioblastoma cells (Supplementary Fig. 6) and assessed for survival in the presence of dimethyl sulfoxide (DMSO), 100 nM DNMDP, or 100 nM trequinsin. After elimination of highly variable sites due to low representation in the PDE3A DMS library (Supplementary Fig. 6), we compared survival results from cells treated with 100 nM DNMDP or DMSO (Fig. 6a).

The majority of mutations that enabled GB1 cell survival mapped to residues contributing to the stability of the protein fold. Remaining loss-of-function mutations that enabled GB1 cell survival in the presence of DNMDP fell largely into two categories: (i) mutations that cluster around or in the catalytic site and (ii) mutations that surround the PDE3A homodimerization domain (Fig. 6b). These mutations permit GB1 survival in the presence of DNMDP, with a $\log_2$ fold change (LFC) >0, similar to that supported by introduction of stop codons. We previously reported that mutations of the PDE3A active site abolished DNMDP binding and cytotoxic response[3]. The DMS and HDX results also implicate the dimerization interface of PDE3A as having a key role in PDE3A-SLFN12 complex formation. Supporting these results, mutation of the PDE3A homodimerization domain, exemplified by N867R, did not prevent compound binding (Fig. 6c) but did inhibit PDE3A-SLFN12 complex formation and DNMDP response (Fig. 6d–f). Interestingly, several substitutions of F914, in a region of PDE3A implicated in SLFN12 binding by HDX and Cryo-EM, also caused resistance to DNMDP. We therefore individually assessed two mutations of F914 and found that F914A and F914D retained the ability to bind resin-conjugated compound[1], albeit with decreased efficiency (Fig. 6c), but no longer complexed with SLFN12 in response to DNMDP (Fig. 6d) and failed to support DNMDP cytotoxic response (Fig. 6e). Mutation of several other PDE3A residues determined by Cryo-EM to be located at the PDE3A–SLFN12 interface also caused resistance to DNMDP (Supplementary Table 4 and Supplementary Data 2).

**SLFN12 has RNase activity, which is required for DNMDP-induced cell killing.** SLFN13 was recently reported to contain an N-terminal RNase domain[20], a region with 35% sequence identity to SLFN12 (Supplementary Fig. 7). We hypothesized that SLFN12 is also an RNase and that this activity may contribute to

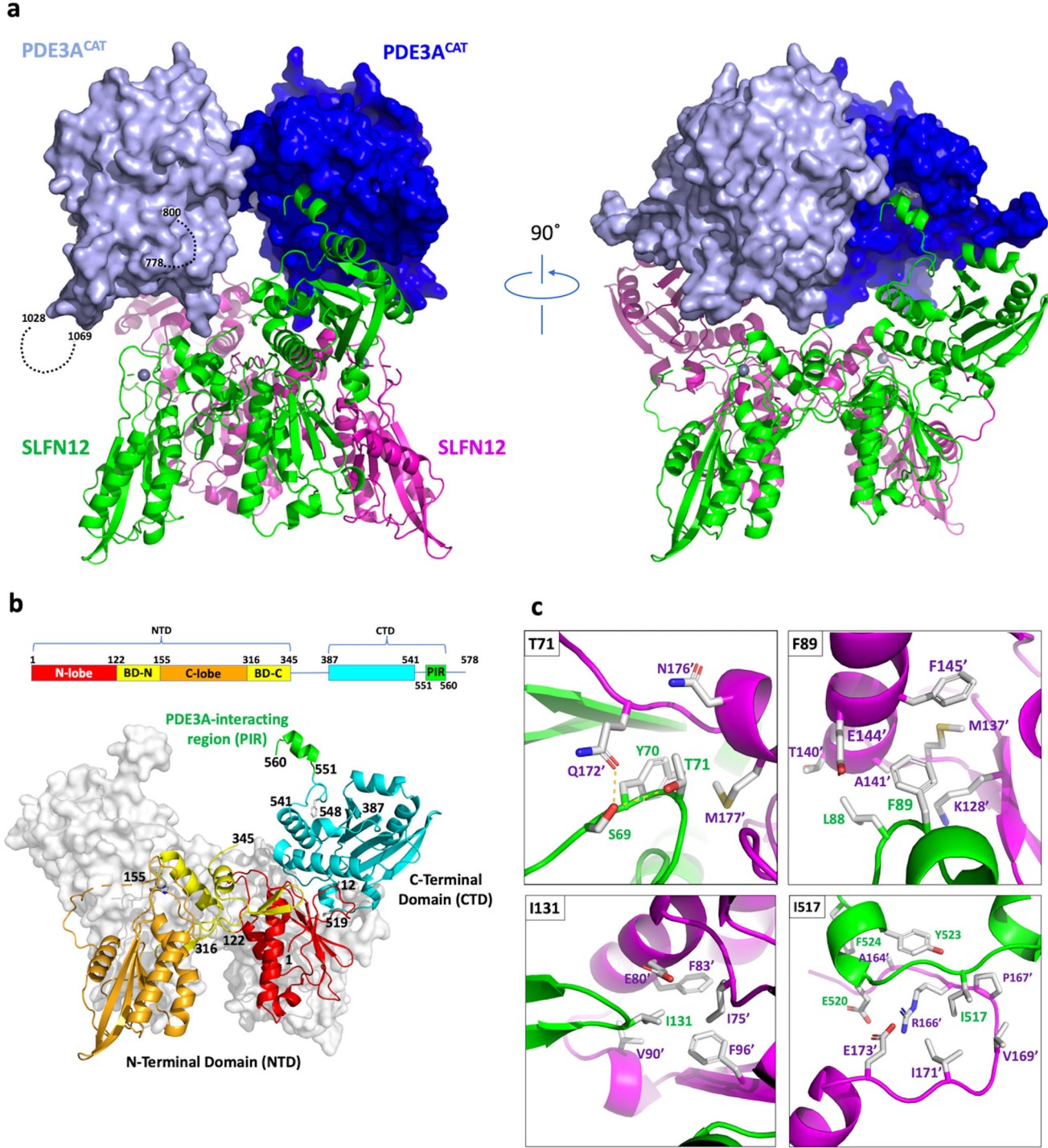

**Fig. 3 Overview of the Cryo-EM structure of the PDE3A$^{CAT}$ and SLFN12 complex. a** Structure of the PDE3A$^{CAT}$-SLFN12-DNMDP heterotetramer. PDE3A$^{CAT}$ (dark and light blue) and SLFN12 (green and magenta) are shown in a surface and cartoon representation, respectively. The loop regions between 779–799 and 1029–1068, which do not show density in the Cryo-EM maps, are indicated by dashed lines on one PDE3A monomer (light blue). The zinc ions of SLFN12 are shown as gray spheres. **b** Detailed view of the SLFN12 monomer structure. A schematic showing the different regions of SLFN12 is shown at the top with the structure of the dimer shown below. One monomer is shown in a surface representation (white) and the other in cartoon representation with different regions color-coded to match the schematic. Residues that are involved in stabilizing contacts within the C-terminal domain and between the N- and C-terminal domains (F548, A12, and P519) are indicated. BD bridging domain. **c** Summary of the SLFN12 dimer interface interactions. Each of the points of contact is denoted by a key residue, which is shown in the top left corner of each panel. The backbone of each monomer, and the labels of the residues shown, are colored green or magenta depending on which monomer they come from. The labels for the monomer in magenta also have a prime mark to indicate that they are from the other monomer.

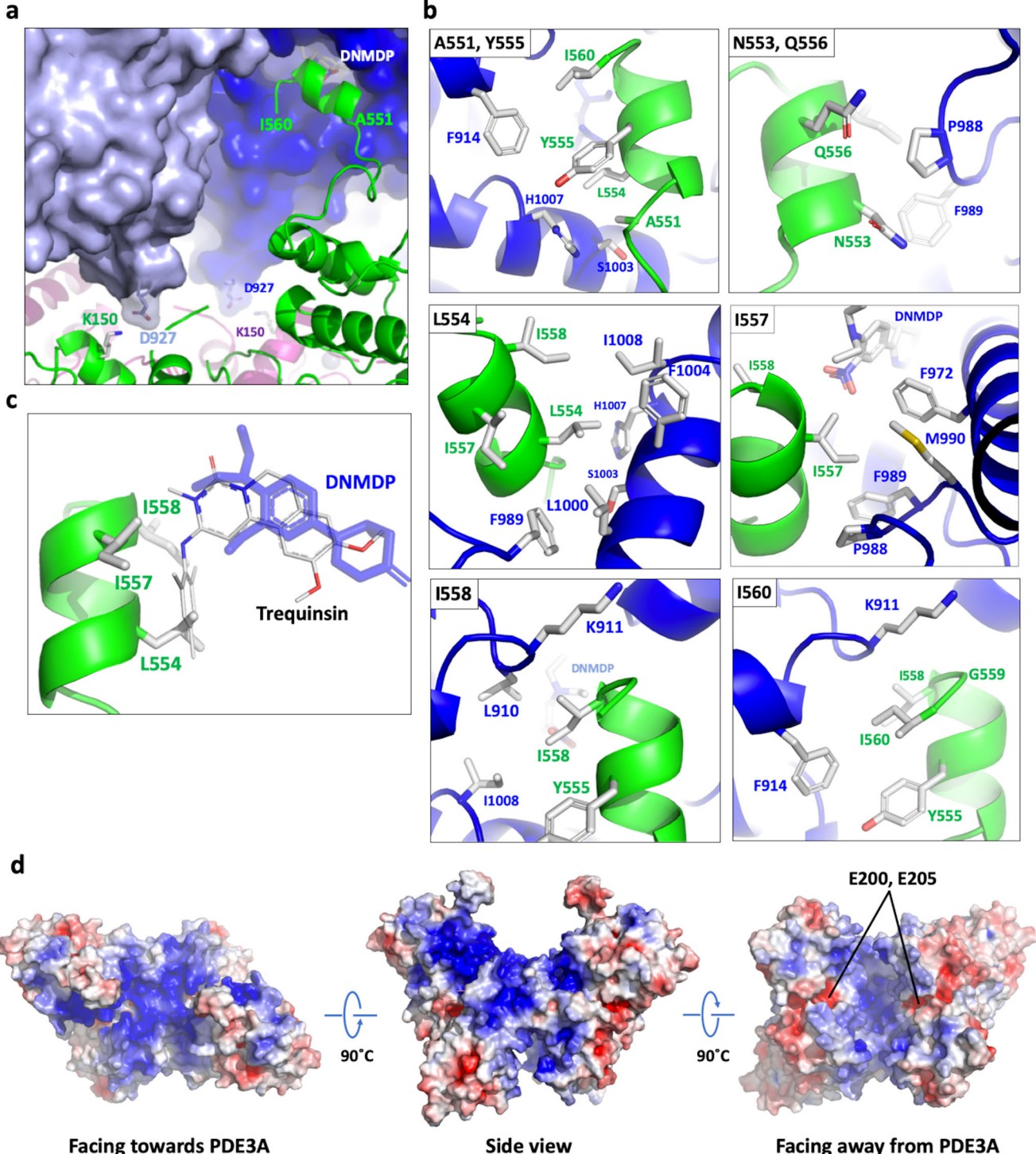

**Fig. 4 Intermolecular interactions between PDE3A^CAT and SLFN12. a** Summary view of intermolecular interactions between PDE3A^CAT and SLFN12. The surface representation around D927 was made transparent to highlight the side chains. The residues are labeled in the color of the protein monomer they come from. **b** Interactions between the C-terminal region of SLFN12 and PDE3A. The SLFN12 residues that are the focus of each panel are shown in the upper left corner. The backbone and labels for the side chains of PDE3A are shown in blue and SLFN12 in green. **c** Modeling the effect of trequinsin on PDE3A^CAT-SLFN12 complex formation. Trequinsin, shown in a thin stick format and colored per atom to differentiate from DNMDP (blue), is modeled based on superimposing the PDE3A^CAT-Xtl-trequinsin structure onto PDE3A^CAT in the Cryo-EM structure. **d** Distribution of surface charges on SLFN12. For the last figure, the location of the two putative RNAse catalytic residues on each monomer are indicated. The surface charge for the SLFN12 dimer was calculated using APBS[47] and visualized in PyMOL. The surface was colored with a gradient going from blue (highly positively charged) to white (neutral), to red (highly negatively charged).

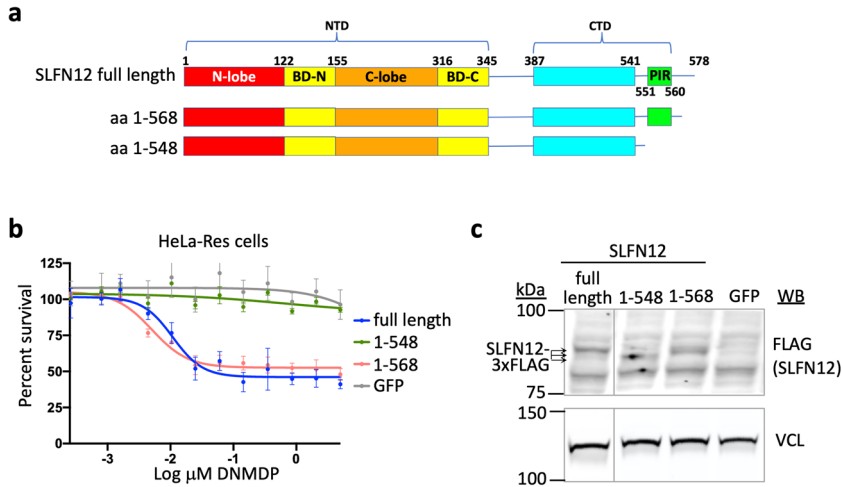

**Fig. 5 The PIR helix of the SLFN12 CTD is required for DNMDP-induced cancer cell killing. a** Schematic diagram of SLFN12 deletion mutants. **b** Seventy-two-hour DNMDP-response viability assay of SLFN12 truncation mutants deleting the PIR region expressed in HeLa-Res cells lacking endogenous SLFN12 expression. Data are plotted as mean values with error bars indicating $+/-$standard deviation of four replicates. **c** Anti-flag immunoblot analysis showing the expression levels of flag-tagged SLFN12 proteins, marked with arrows.

DNMDP-induced cell killing. To test our hypothesis, we mutated two residues of SLFN12, E200 and E205, orthologous to the active site residues of SLFN13 (Supplementary Figs. 5 and 7), and analyzed the effect of these mutations on DNMDP-induced complex formation and cell killing. The SLFN12 mutants all interacted with endogenous PDE3A in HeLa cells upon treatment with DNMDP (Fig. 7a), suggesting that the mutations did not interfere with complex formation. However, whereas ectopic expression of wild-type SLFN12 conferred DNMDP sensitivity in HeLa-Res cells lacking endogenous SLFN12 expression, ectopic expression of mutant SLFN12 did not, suggesting a requirement for an E200/E205-dependent SLFN12-intrinsic enzymatic activity for response to DNMDP (Fig. 7b).

To determine whether SLFN12 is indeed an RNase, we incubated 2 µM recombinant SLFN12 with human rRNA isolated from HeLa cells in a buffer containing 40 mM Tris-HCl, pH 8.0, 20 mM KCl, 4 mM $MgCl_2$, and 2 mM dithiothreitol (DTT). At this protein concentration, most of the SLFN12 protein would be expected to exist in a dimeric form. Under these conditions, wild-type SLFN12, but not heat-denatured SLFN12, was able to degrade the rRNA (Fig. 7c). This RNase activity was largely inhibited by mutation of E200 or E205. We repeated this experiment with 0.25 µM SLFN12, a concentration at which the SLFN12 is predicted to be predominantly monomeric (Fig. 7d), and found that SLFN12 did not affect the integrity of the substrate rRNA by itself. Addition of PDE3A^CAT resulted in a measurable decrease in rRNA integrity, and further addition of DNMDP or estradiol[6], but not trequinsin, greatly enhanced rRNA degradation. Consistent with a requirement for the C-terminal helix of SLFN12 in binding to PDE3A, no activation of SLFN12 RNase activity was observed upon incubation of the isolated SLFN12 NTD with DNMDP-bound PDE3A (Fig. 7e). Taken together, these results support the hypothesis that SLFN12 encodes an RNase, that SLFN12 RNase activity is stimulated by DNMDP-induced complex formation with PDE3A, and that SLFN12 RNase activity is required for DNMDP-induced cancer cell death.

## Discussion
We have performed a comprehensive structure–function analysis of the DNMDP-induced PDE3A-SLFN12 complex that provides significant insight into how the complex forms, the role of

DNMDP in stabilizing the complex, and the function of SLFN12 in cellular response to DNMDP. DNMDP binds the active site of PDE3A, forming a surface for high-affinity interaction of SLFN12. Because DNMDP-bound PDE3A forms an adhesive surface for SLFN12, we propose to name this class of compounds "velcrins". The set of known velcrins currently includes DNMDP, zardaverine, BRD9500, estradiol, anagrelide, nauclefine, and several progesterone receptor agonists[1,2,4–7,26]. We propose that velcrins promote dimerization of SLFN12 by recruiting two SLFN12 monomers to a constitutive PDE3A dimer, with the second SLFN12 monomer likely binding cooperatively. We hypothesize that dimerization of SLFN12 may stimulate SLFN12 RNase activity, analogously to activation of RNaseL by 2'–5'-linked oligoadenylates[27]. Dimerization of SLFN12 likely further stabilizes the PDE3A-SLFN12 complex, as the HDX data identified reduced deuterium uptake at the PDE3A homodimerization interface when SLFN12 was present.

The Cryo-EM structure of the PDE3A^CAT-SLFN12 complex revealed that the majority of contacts between the two proteins are made by a single α-helix comprised of 10 amino acids in the C-terminal region of SLFN12 in each monomer. Residues of this C-terminal helix make multiple contacts to PDE3A, including to PDE3A F914, an amino acid determined by DMS to be essential for complex formation and velcrin response. No other human SLFN family members share the primary sequence found in this region of SLFN12, perhaps explaining why no other SLFNs have been found to complex with PDE3A, with or without velcrin treatment. DNMDP provides additional contacts to residues in SLFN12 that stabilize the complex. However, in the case of PDE3A-bound trequinsin, a direct steric clash with residues in the C-terminal region of SLFN12 prevents complex formation. The biophysical and structural studies also implicate electrostatic interactions as important for stability of the PDE3A-SLFN12 complex.

*SLFN11*, *SLFN13*, and *SLFN14* all encode an RNase activity[18–20] and have been shown to play a role in restriction of viral infection[20,28,29]. We speculate that SLFN12 may have a similar physiological function that is leveraged by velcrins to kill cancer cells expressing elevated levels of PDE3A and SLFN12. Velcrins have been shown to synergize with the anti-viral, cell death-inducing interferons[4], and another component of the anti-viral innate immune response, RNase L, is able to mediate killing of virally infected cells[30–32], establishing a precedent for cytotoxic

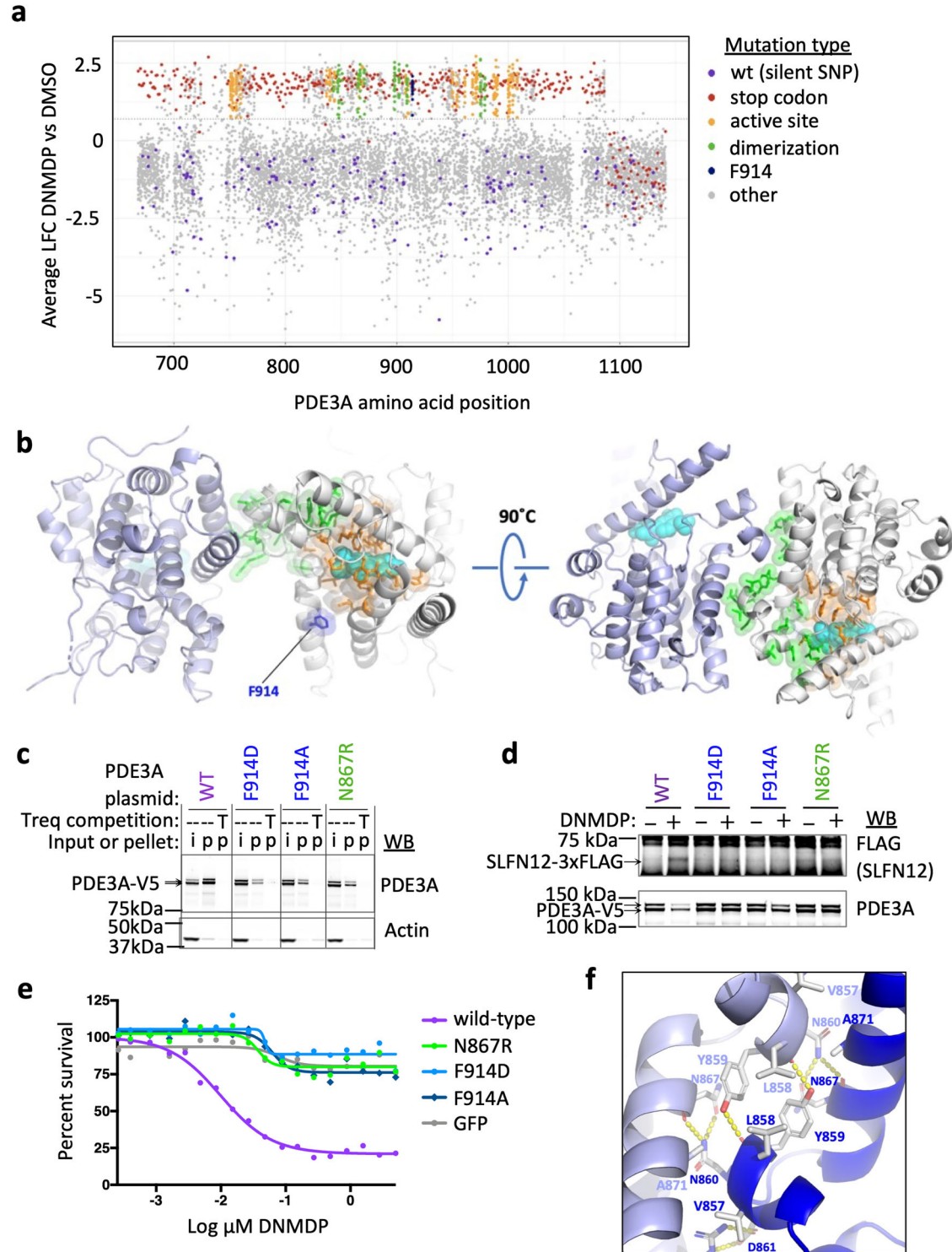

response to RNase activation. Here we demonstrate that SLFN12 is also an RNase, that this RNase activity is increased by incubation with velcrin-bound PDE3A, and that SLFN12 RNase activity is essential for velcrin-induced cancer cell killing.

Unlike traditional targeted therapies that leverage dependencies created by genomic alterations in cancer cells, velcrins instead induce cancer cell death by a novel gain-of-function mechanism mediated by PDE3A-SLFN12 complex formation. Inhibition of PDE3A enzymatic activity does not correlate with cancer cell killing caused by PDE3A-SLFN12 complex formation[1] and may not be required[26]. PDE3A-SLFN12 complex formation is thus the

critical event in velcrin response, and the structure reported here provides the molecular details of this interaction. Further development of our understanding of the mechanism of action of the velcrin-induced PDE3A-SLFN12 complex in cancer cell killing will support evaluation of velcrins as potential cancer therapeutics.

## Methods

**PDE enzyme assay**. PDE activity assays were performed as previously described[33]. Briefly, cell-free extracts were prepared from Sf9 insect cells recombinantly expressing full-length human PDEs (PDE2A, 3A, 3B, 4B, 7B, 8A, 9A, 10A, 11A).

**Fig. 6 Deep mutational scanning of PDE3A identifies DNMDP resistance mutations. a** Average log2 fold changes (LFC) in abundance of mutant alleles of PDE3A following treatment with DNMDP as compared to treatment with DMSO. Wild-type (wt) alleles are marked with silent single-nucleotide polymorphisms (SNPs) (purple), whereas stop codons (red) result in complete loss of function until the last 59 amino acids. One standard deviation of increased survival is indicated with the dotted line. Resistance mutations with a survival score greater than one standard deviation from the mean and that are not predicted to disrupt PDE3A protein folding are colored to indicate localization to the active site (orange), homodimerization domain (green), or putative binding site of SLFN12 (dark blue). **b** Location of resistance mutations mapping to the PDE3A active site (orange), PDE3A homodimer interface (green), and putative SLFN12-binding site (dark blue). The backbone of the two PDE3A monomers are shown in light blue and white, with the mutations mapped onto the white monomer. The DNMDP is represented as space-filling and colored cyan. **c** Pulldown of wild-type or mutant PDE3A with a resin-conjugated DNMDP analog following ectopic expression in PDE3A-knockout A2058 cells. PDE3A proteins (marked with arrows) recovered in the pellet (P lanes) were visualized by western blotting using a PDE3A antibody. The presence of 10 μM trequinsin (T lanes) during pulldown competed away the binding of PDE3A to the resin, resulting in a negative pulldown. **d** Co-immunoprecipitation of ectopically expressed wild-type or mutant PDE3A with flag-tagged wild-type SLFN12 following ectopic expression in PDE3A-knockout HeLa cells. SLFN12 proteins, marked with arrows, that co-immunoprecipitated with PDE3A were detected by anti-FLAG western blotting. **e** Confirmation of deep mutation scanning results with selected PDE3A mutants in PDE3A-knockout A2058 cells. Seventy-two-hour Cell-Titer Glo assay. GFP, green fluorescent protein. **f** Dimer interface in the PDE3A$^{CAT-Xtl}$ structure. The labels for the residues are colored light and dark blue depending on which monomer they come from.

PDE5 was purified from human platelets by homogenization (Microfluidizer, 800 bar, 3 passages), centrifugation (75,000 × $g$, 60 min, 4 °C), and ion exchange chromatography of the supernatant on a Mono Q 10/10 column (linear NaCl gradient, eluted with 0.2–0.3 M NaCl in a buffer containing 20 mM Hepes pH 7.2 and 2 mM MgCl$_2$). Fractions containing PDE5 activity were pooled and stored at −80 °C. PDE6 was purified from rod outer segments of bovine retinae by Dr. Körschen, Forschungszentrum Jülich, Germany[34]. Bovine PDE1 was purchased from Sigma-Aldrich (P9529, Taufkirchen, Germany). Enzyme inhibition studies were carried out using the commercially available 3H-cAMP and 3H-cGMP Scintillation Proximity Assay (SPA) systems (RPNQ0150, Perkin-Elmer, Rodgau, Germany).

**Cloning, expression, and purification of the catalytic domain constructs of PDE3A.** For the biophysical studies, the region of PDE3A that codes for residues 640–1141, referred to as PDE3A$^{CAT}$, was codon optimized (Supplementary Table 6) for *Escherichia coli* expression (GeneArt, Thermo Fisher Scientific) and inserted into an expression vector with a pET21 backbone that was modified to append an N-terminal polyhistidine sequence, a Strep-tag sequence, and a TEV protease cleavage site. The vector was transformed into One Shot BL21 (DE3) cells (Thermo Fisher Scientific) and then grown at 37 °C until mid-log phase. At this point, the temperature was reduced to 18 °C, expression was induced with IPTG (Teknova), and the cells were left to grow overnight. The cells were resuspended in the following lysis buffer: 50 mM Tris pH 8, 500 mM NaCl, 1 mM MgCl$_2$, 0.5 mM TCEP, 20 mM imidazole, complete EDTA-free protease inhibitors (Roche), and lysozome (Millipore). The resuspended cells were lysed using an Emulsiflex-C3 homogeniser (Avestin), and the protein was affinity purified using a Ni-charged HisTrap HP column (GE Healthcare). The running buffers for the HisTrap were as follows: HisTrap-Buffer A was comprised of 50 mM Tris pH 8, 500 mM NaCl, and 1 mM MgCl$_2$ and HisTrap-Buffer B contained the same components and included 1 M imidazole. After HisTrap purification, the protein was passed over a Superdex 200 26/600 size exclusion column (GE Healthcare) equilibrated with the SEC-Buffer: 50 mM Tris pH 8, 500 mM NaCl, 1 mM MgCl$_2$, and 0.5 mM TCEP. The affinity tag was cleaved by addition of poly-histidine-tagged TEV protease and incubating overnight at 4 °C. The TEV protease was removed along with the tag by passing over the Ni-charged HisTrap column. The catalytic domain construct used for crystallography, referred to as PDE3A$^{CAT-Xtl}$, comprised residues 669–1095 with replacement of two internal loops with shorter GGSGGS linkers. Specifically, the construct comprised residues 669–779 followed by a GGSGGS linker, residues 801–1028, followed by a GGSGGS linker, and residues 1068–1095. The protein was expressed and purified as described. For generation of biotinylated protein, the gene for PDE3A$^{CAT}$ was inserted into a similar expression vector as described but which appended an Avi-tag sequence to the C-terminus. The expression and purification were performed in the same manner as described for the other constructs. The Avi-tagged protein was biotinylated using BirA ligase and the excess biotin removed by size exclusion. All of the proteins were dialyzed overnight into 20 mM Hepes pH 7, 150 mM NaCl, 1 mM MgCl$_2$, 0.5 mM TCEP, and 10% glycerol, concentrated, aliquoted, and stored at −80 °C until use.

**Cloning, expression, and purification of SLFN12.** Full-length SLFN12 was codon-optimized for insect expression (Supplementary Table 6), and the cDNA was synthesized at GeneArt. The gene was inserted into a modified pFastBac vector that was designed to append an N-terminal affinity tag comprised of a polyhistidine sequence, the MBP, and a TEV protease cleavage site. The vector was transformed into MAX Efficiency DH10Bac competent cells (Thermo Fisher Scientific), plated, and a colony from a re-streaked plate used for an overnight growth to generate a bacmid stock. The bacmid was transfected into sf9 cells (Expression Systems) using Cellfectin II (Thermo Fisher Scientific), and the cells were grown at 27 °C in a shaker incubator for 3 days with ESF921 cell culture media (Expression Systems).

The supernatant of the infected cells was used to infect further sf9 cells to generate a stock of baculovirus-infected insect cells (BIICs), which were aliquoted and stored in a liquid N$_2$ freezer. Large-scale preps were performed with a 1:1000 infection ratio of BIICs to media and incubating the cells for 3 days at 27 °C in a shaker incubator. The cells were resuspended in the lysis buffer described, lysed using an Emulsiflex-C3 homogeniser, clarified by centrifugation, and the protein affinity purified over a Ni-HisTrap HP column followed by an MBPTrap HP column (GE Healthcare). The MBPTrap was equilibrated with HisTrap-Buffer A and the protein eluted by supplementing this buffer with 10 mM maltose. The MBP tag was cleaved with TEV protease and removed upon passing over a Superdex 200 26/600 column in series with an MBPTrap column. The protein was dialyzed overnight into 20 mM Hepes pH 7.4, 500 mM NaCl, 500 μM TCEP, and 10% glycerol, concentrated, aliquoted, and stored at −80 °C until use. In all, 500 mM NaCl was kept in the purification buffer throughout due to a propensity of SLFN12 to aggregate at high protein concentration and low NaCl concentration.

**SEC and MALS analysis.** The SEC and SEC-MALS analysis of complex formation of the wild-type PDE3A$^{CAT}$ and SLFN12 was performed with a WTC-030S5 (Wyatt technologies Inc.) size exclusion column attached to an Agilent 1260 Infinity HPLC. The system was equilibrated in 50 mM Tris pH 8.5, 150 mM or 500 mM NaCl, and 1 mM MgCl$_2$. Protein(s) were diluted in this running buffer to a final concentration of 10 μM. Where relevant, DNMDP was added to a final concentration of 100 μM. Complexes were incubated for at least 20 min before loading 50 μl onto the column, which was run at a flow rate of 0.4 ml/min. Light scattering data were collected on a MiniDAWN TREOS (Wyatt Technologies Inc.) connected in series after the ultraviolet (UV) module of the HPLC. The SEC-MALS analysis for SLFN12 was only performed at 500 mM NaCl as a well-defined peak was not observed at 150 mM NaCl. The SEC-MALS analysis for PDE3A + SLFN12 was performed at 150 mM NaCl as these conditions gave a single well-defined peak. PDE3A$^{CAT}$ gave essentially the same result for 150 and 500 mM NaCl. At 500 mM NaCl and pH 8.5, SLFN12 gave a well-defined peak, albeit eluting at a much later elution volume than would be expected for a protein this size. At pH 7.4 or 150 mM NaCl, SLFN12 did not produce a well-defined peak. It is not clear the cause of these effects, but it suggests there is a non-specific interaction with the column resin.

**Sedimentation equilibrium analytical ultracentrifugation.** All experiments were performed using a ProteomeLab XL-1 ultracentrifuge (Beckman Instruments). Proteins were dialyzed at 4 °C prior to each experiment in buffer containing 20 mM Hepes (pH 7.4), 500 mM NaCl, 1 mM MgCl$_2$, and 0.1 mM TCEP. SE-AUC experiments were conducted for SLFN12 (0.1 and 3 μM) and PDE3a (0.2, 0.8, and 3.6 μM) at 4 °C in an An-50 Ti rotor at 5152 × $g$ (8000 rpm), 19,340 × $g$ (15,500 rpm), and 54,415 × $g$ (26,000 rpm). Absorbance data were collected at 230 and 280 nm. SedPHAT was used to fit the data to a monomer/dimer self-association model[35,36]. SedNTerp was used to calculate protein and buffer parameters. Images were generated using GUSSI[37].

**Amylose resin pulldown experiments with MBP-SLFN12.** The pulldown experiment was performed with the following reaction buffer: 20 mM Hepes pH 7.4, 1 mM MgCl$_2$, 150 mM or 500 mM NaCl, 100 μM TCEP, and 1% DMSO. When the reaction was performed in the presence of DNMDP or trequinsin, the reaction buffer contained 10 μM DNMDP or trequinsin instead of DMSO. Dilution of the compound from its stock solution left a final DMSO concentration of 1%. In all, 2 μM of His$_6$-MBP-SLFN12 was incubated with 2 μM of PDE3A$^{CAT}$, wild type or mutant, at 4 °C prior to binding to amylose resin (New England Biolabs). After washing the resin with reaction buffer to remove unbound protein, the resin was eluted with 20 mM Maltose. The eluted protein was visualized using sodium dodecyl sulfate-polyacrylamide gel electrophoresis (SDS-PAGE).

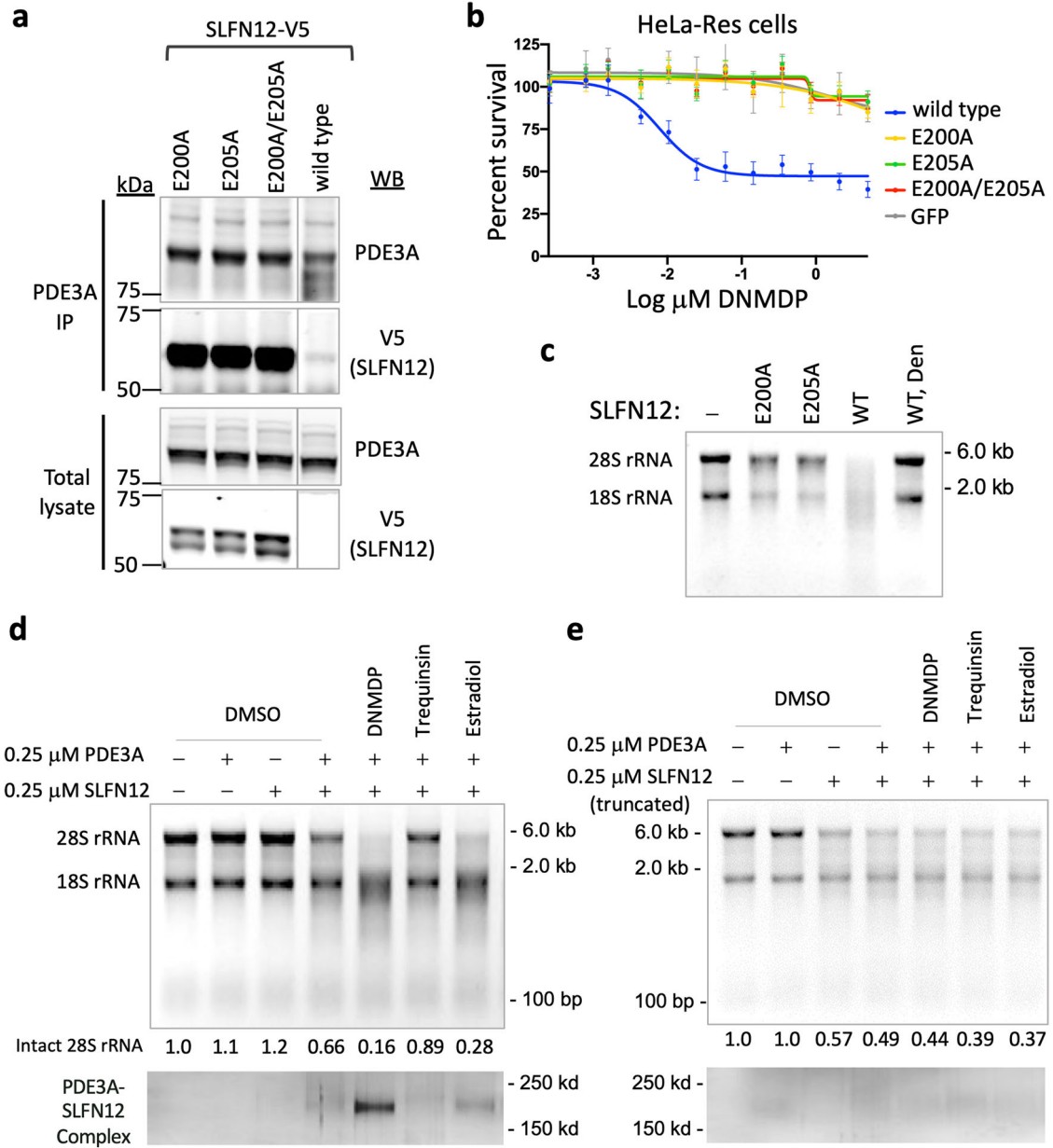

**Fig. 7 SLFN12 RNase activity is required for sensitivity to DNMDP. a** Complex formation of PDE3A with predicted SLFN12 catalytic mutants was assessed by ectopically expressing V5-tagged wild-type or mutant SLFN12 constructs in HeLa cells, treating with DNMDP for 8 h, immunoprecipitating endogenous PDE3A, and detecting coprecipitated SLFN12 by V5 immunoblot. **b** DNMDP sensitivity of SLFN12 cDNAs with mutations of predicted RNase catalytic residues in HeLa-res cells lacking endogenous SLFN12 expression. Data are plotted as mean values with error bars indicating $+/-$standard deviation of four replicates. **c** Direct measurement of SLFN12 RNase activity. In all, 2 μM of wild-type (WT) or the active site mutant (E200A, E205A) SLFN12 were incubated with 2 μg of human rRNA at 37 °C for 40 min. Wild-type SLFN12 was denatured (WT, Den) by heating at 65 °C for 20 min. Cleaved rRNAs were analyzed on a formaldehyde agarose gel. **d** Induction of SLFN12 RNase activity by PDE3A-SLFN12 complex formation. 0.25 μM PDE3A and SLFN12 proteins were incubated with DMSO, 12.5 μM DNMDP, 12.5 μM trequinsin, or 12.5 μM estradiol at room temperature for 30 min prior to a 1:10 dilution of complex and incubation with rRNA. Digested rRNAs were analyzed on a denaturing agarose gel. The relative amount of intact 28S rRNA was quantified using the ImageJ software, shown at the bottom of the figure. To detect the complex formation between PDE3A and SLFN12, preincubated recombinant proteins were crosslinked using 1-ethyl-3-(3-dimethylaminopropyl)carbodiimide (EDC) and *N*-hydroxysuccinimide (NHS). Reaction was stopped by adding SDS loading buffer. Proteins were separated on an SDS-PAGE gel and detected by silver staining. **e** RNase activity of the isolated SLFN12 NTD. Experiment with SLFN12 amino acids 1–347 performed as in **d**.

**BLI analysis of complex formation.** The BLI data were collected on an Octet Red384 system (ForteBio, Sartorius). Biotinylated PDE3A(640-1141) was immobilized on streptavidin biosensors (ForteBio, Sartorius) by dipping the sensors into 1 ng/μl PDE3A in 20 mM Hepes pH 7.4, 500 mM NaCl, 100 μM TCEP, 5 mM MgCl₂, 0.05% P20, and 0.1% DMSO for 60 s. The sensors were then dipped into a well containing 0, 12.5, 25, 50, 100, 200, 400, and 800 nM SLFN12 in the same buffer for 120 s to obtain the association step. The subsequent dissociation step was performed with buffer alone for 300 s. For the effect of the compounds on complex formation, the experiments were repeated with the buffer including 10 μM DNMDP or trequinsin in the reaction buffer. The steady-state data were fit using the GraphPad Prism 7 software assuming a 1:1 stoichiometry of SLFN12 to PDE3A.

**Differential scanning calorimetry (DSC) analysis of the catalytic domain of PDE3A.** PDE3A^CAT was dialyzed overnight into the DSC buffer, which was composed of 20 mM Hepes pH 7.4, 150 mM NaCl, 1 mM MgCl₂, and 100 μM TCEP. Melting temperature data were collected using a Microcal VP Capillary DSC (Malvern Panalytical). In all, 4 μM PDE3A^CAT was incubated in the absence or

presence of 10 μM DNMDP for 10–20 min prior to data collection. A similar approach was taken with SLFN12 but with 500 mM NaCl in the DSC buffer. Data were collected from 20 to 90 °C, with a 200 °C/h scan rate and 8 s filtering period, and were analyzed in the Origin software.

**Structure determination of the catalytic domain of PDE3A.** The *E. coli* expressed PDE3A protein was dialyzed overnight into 20 mM Hepes pH 7.4, 50 mM NaCl, 1 mM MgCl$_2$, and 0.5 mM TCEP and then concentrated to 10 mg/ml. The protein was mixed in a 1:2 protein:well ratio with well conditions of 100 mM MES pH 5.7–6.1, 100 mM calcium acetate, and 16–18% PEG 3350, and crystals were grown using hanging drop vapor diffusion crystal trays (QIAGEN). Plate clusters appeared within 2 days, which were used to streak seed further drops containing the same conditions but reduced PEG 3350 concentrations of 12–14%. Single crystals grew within a day. For data collection, the crystals were transferred to drops that contained an increased PEG 3350 concentration of 30% and either no compound or 100 μM DNMDP, trequinsin, or cAMP. After incubation for 10 min, the crystals were flash frozen. Crystallographic data were collected by Helix BioStructures LLC at the ESRF, beamline ID-30B. The diffraction data were processed with XDS[38]. Molecular replacement, and subsequent refinement, was performed in Phenix[39], using the catalytic domain of PDE3B[23] (PDB accession code:1SO2) as the initial search model. The final refined structure has four molecules in the asymmetric unit cell, corresponding to two dimers. The active site of each of the four protein monomers contains clear density for DNMDP and trequinsin, where relevant, along with density for two hexacoordinated metal ions. Initially, the metal ions were refined as Mg$^{2+}$. While one Mg$^{2+}$ refined with *B*-factors that matched surrounding atoms and showed no additional density, the second putative Mg$^{2+}$ ion exhibited a lower-than-expected *B*-factor and positive electron density. This atom was subsequently identified to be Mn$^{2+}$ based on analysis of anomalous electron density maps[40].

**Analysis of the PDE3A/SLFN12 interaction interface by HDX-MS.** For HDX-MS experiments, PDE3A$^{CAT}$ (22 μM in 20 mM Hepes, 500 mM NaCl, 100 μM TCEP, 1 mM MgCl$_2$, pH 7.4) was incubated with BRD9500 (40 μM) for 30 min at room temperature to ensure complex formation, and the complex was subsequently analyzed either alone or in the presence of SLFN12 (22 μM in above buffer). The complex was diluted tenfold with deuterated Hepes buffer, resulting in 99.5% of PDE3A$^{CAT}$ molecules bound to BRD9500 during labeling. For the analysis of PDE3A$^{CAT}$:BRD9500 in complex with SLFN12, the proteins were mixed in an equimolar concentration (22 μM) and diluted as described above, resulting in ~89% of protein molecules in a bound state for both PDE3A$^{CAT}$ and SLFN12 during labeling. Samples were quenched with ice-chilled guanidine hydrochloride (4 M, 1.25% formic acid) in a ratio of quench:sample 3:2 (v:v) so that the final pH was 2.5 (0.75% formic acid).

Labeling experiments were performed at 20 °C in triplicates and labeled samples were quenched at 30, 300, and 3000 s. For peptide identification experiments, samples were diluted with protiated Hepes and quenched as described above. Samples were injected to the MS via an automated HDx-3 PAL™ system (LEAP Technologies, Morrisville, NC). Samples were incubated in the loop for 1 min and were subsequently digested online for 3 min at 300 μl/min using an immobilized protease type XIII/pepsin column (w/w, 1:1, NBA2014002, NovaBioAssays, Woburn, MA), operated at 8 °C. Peptides were trapped onto an Acclaim™ PepMap™ 300 μ-Precolumn™ (C18, 1 × 15 mm, 163593, Thermo Fisher Scientific, Waltham, MA) using solvent A (0.1% formic acid (v/v)) and were separated onto a Hypersil Gold C18 (1 × 50 mm, 1.9 μM, 25002–051030, Thermo Scientific, Waltham, MA) at 40 μl/min using solvents A and B (0.1% v/v formic acid in acetonitrile). The following gradient was applied: 3% B to 10% B in 0.1 min, to 35% B in 15 min, to 95% B in 2 min; kept at 95% B for 1 min and returned to initial conditions in 0.9 min. Both the trap and analytical column were operated at 0 °C.

MS analysis was carried out on an Orbitrap Fusion™ Lumos™ Tribrid™ Mass Spectrometer (Thermo Fisher Scientific) using a spray voltage of 3.5 kV, capillary temperature 220 °C, and vaporizer temperature 50 °C. Full MS scans were acquired in the *m/z* range 300–1300, with an AGC target 5e5 and 60,000 resolution (at *m/z* 200). Peptides with charge states 2–6 were selected for MS/MS fragmentation using higher-energy C-trap dissociation with an AGC target 4e5, loop count 12, and isolation window 2 *m/z*. Data were acquired in profile mode and peptides were identified using Spectrum Mill Proteomics Workbench (prerelease version B.06.01.202, Agilent Technologies). Searches were performed using ESI QExactive in the Instrument menu with All in the fragmentation mode. A non-specific enzyme search was performed using a SwissProt FASTA file containing common contaminants and sequences of pepsin and protease type XIII. Peptide and fragment tolerances were at ±20 ppm and peptide false discovery rate at 1%. Further processing, manual validation of the data, and deuterium uptake measurements of PDE3A with or without SLFN12 occurred in HDExaminer (Sierra Analytics). Differences in the PDE3A deuterium levels >4% were considered significant. The data have not been corrected for deuterium back exchange, following suggestions in the HDX community recommendations paper (Masson et al.[24]).

**Cryo-EM sample preparation and data acquisition.** PDE3A$^{CAT}$, SLFN12, and DNMDP were incubated in a 1:1:1 molar ratio and passed over a Superose 6 10/300 equilibrated with 20 mM Hepes pH 7.4, 150 mM NaCl, 1 mM MgCl$_2$, and 0.5 mM TCEP. The complex was concentrated and flash frozen until use. SEC analysis of a thawed sample of the complex did not show any aggregation. The complex was diluted to 0.6 mg/ml in the storage buffer and supplemented with 0.0038% NP-40s detergent immediately prior to freezing. CFlat 1.2/1.3 400 Au mesh grids (Protochips, Inc.) were glow discharged for 45 s using ambient air in a Gatan Solarus plasma cleaner (Gatan, Inc.). Grid freezing was performed using a Vitrobot Mk IV (ThermoFisher, Inc.), with the blotting chamber held at 100% humidity and 18 °C. In all, 3.5 μl of sample was applied to a grid, blotted immediately for 4.5 s, and then immediately plunge frozen in liquid ethane. Data were collected from the grid using a Titan Krios (ThermoFisher, Inc.) electron microscope equipped with a K3 camera and energy filter (Gatan, Inc.). The microscope was operated at a nominal magnification of ×81,000 (pixel size 1.063 Å) with the camera in super-resolution mode. Each micrograph encompassed a total dose of 50 e$^-$/Å$^2$ to the sample, fractionated over 40 movie frames.

**Cryo-EM data processing.** Micrograph movies were summed and dose-weighted using Relion 3.1[41]. Contrast transfer function (CTF) estimation for each was performed using CTFFind 4.1.13[42] on power spectra from movie frame sums over a 4 e$^-$/Å$^2$ dose. Exposures that were outliers in fitted motion or CTF parameters were removed, resulting in a total 2393 micrographs used for downstream data processing. The particle processing steps outlined here are also presented in Supplementary Fig. 5. Using CisTEM 1.0.0 beta[43], 2,160,426 potential particle positions were picked and filtered over 2 rounds of two-dimensional classification. The resulting 408,690 particles were used in CisTEM to generate an ab initio three-dimensional (3D) model and a preliminary 3D auto-refinement. Refined particle positions were then extracted in Relion 3.1, which was used for all subsequent steps. All steps performed in Relion after an initial 3D refinement utilized a solvent mask generated from the previous refinement. 3D classification with full angular and translational searches was used to remove remaining junk particles. The resulting 343,715 particles underwent 2 cycles of 3D refinement and CTF parameter refinement. The resulting 3D refinement was used as the basis for per-particle motion correction (Bayesian polishing). Polished particles were subjected to 3D classification using fixed particle poses to further remove outlier conformations. The resulting 247,402 particles underwent 3 cycles of 3D refinement coupled to CTF refinement, in which all relevant parameters were fit (per-particle defocus, per-micrograph astigmatism and *B*-factor, beam tilt, trefoil, and fourth-order aberration). The resulting 3D refinement was used as the basis for multi-body refinement, splitting the complex into the SLFN12 and PDE3A halves. A 47,632-particle subset was selected based on particles with eigenvalues between ±1.0 along the first eigenvector in the multi-body refinement, and these were used for a final consensus 3D refinement. C2 symmetry was used for all 3D refinements, except those used as inputs to 3D classification.

**Cryo-EM model building.** An atomic model for each dimer pair of the complex was built manually into the two multi-body maps using Coot and refined by iterating between automated real-space refinement using Phenix[44] and manual editing. Completed models for the SLN12 and PDE3A bodies were merged by rigid body docking into the multi-body subset consensus map using Chimera[45]. C2 symmetry was enforced for the model during the automated refinement and docking steps. For model building and refinement, maps were sharpened by an automatically determined *B*-factor followed by filtering to local resolution using Relion 3.1. Reported map/model fit values (FSC, masked cross-correlation, EMRinger score[46]) were calculated based on unsharpened maps. Group (per-residue) model atomic displacement factors were fit during refinement against the sharpened/filtered multi-body maps in Phenix.

**DMS expression library design and production.** Lentiviral vector pMT_BRD025 was developed by the Broad Institute Genetic Perturbation Platform (GPP). Open reading frames (ORFs) can be cloned in through restriction/ligation at a multiple cloning site. The ORF expression is driven by EF1a promoter. A *PAC* gene is driven by SV40 promoter to confer puromycin resistance. Full-length wild-type PDE3A cDNA with six potential PAM sites mutated to synonymous codons was synthesized and cloned into pMT_BRD025 using restriction fragment cloning.

We designed mutations of every codon in the C-terminal region of PDE3A, from codon 668 to codon 1141. At each codon position (668–1141), we tried to make 19 missense changes and 1 nonsense change. Due to the constraint of avoiding specific restriction enzyme sites inside of the ORF sequence, in rare cases some intended codon changes were not possible; there were seven variants missing in the designed library. In addition, we incorporated 188 silent changes along the C-terminal region of interest. It is important to note that, in our design, we minimized use of codons that differ from the corresponding wild-type template codon by a single nucleotide. In all, this library included 9661 variants.

The mutagenesis library we designed was synthesized by Twist BioScience. The library was delivered as a pool of linear fragments representing the full-length PDE3A ORF with a short flank sequence at each end. The two flank sequences were designed to facilitate restriction/ligation cloning of the linear fragment library into the pMT_BRD025 expression vector. The linear fragment library and the vector were each digested with NheI and BamH1 and then ligated. The ligation products were transformed into Stbl4 competent cells (New England BioLabs). The

colonies were harvested and the plasmid DNA was extracted via the Maxi Preparation Kit (Qiagen). The resulting plasmid DNA library was sequenced via Illumina Nextera XT platform. The distribution of variants was assessed. Initial analysis of the library indicated representation of 92.6% of intended substitutions.

Lentivirus was produced by the Broad Institute GPP. The detailed protocol is available at http://www.broadinstitute.org/rnai/public/resources/protocols/. Briefly, HEK293T viral packaging cells were transfected with pDNA library, a packaging plasmid containing the *gag*, *pol*, and *rev* genes (e.g., psPAX2, Addgene), and the VSV-G expressing envelop plasmid (e.g., pMD2.G, Addgene), using TransIT-LT1 transfection reagent (Mirus Bio). Media was changed 6–8 h post-transfection. Virus was harvested 30 h post-transfection.

**DMS assay development.** PDE3A was knocked out of the DNMDP-sensitive human glioblastoma cell line, GB1, using guide RNA sequence GTGGCAGACC ATATTTCCCAA. GB1 cells, grown in Eagle's Minimum Essential Medium + 10% fetal bovine serum (FBS), were transduced with a Cas9- and sgRNA-expressing plasmid, selected for 3 days in 1 μg/ml puromycin, and single cell cloned by flow cytometry. The targeted region of PDE3A was PCR-amplified from each clone and deep sequenced (CRISPR sequencing, MGH DNA core, https://dnacore.mgh. harvard.edu/) to identify clones in which all four alleles of PDE3A were disrupted, resulting in no detectable PDE3A protein expression by immunoblotting with an anti-PDE3A antibody (Bethyl A302-741, 1:1000). Clone 6F12 was selected. DNMDP treatment time was optimized using plasmids expressing wild-type PDE3A (pMT_BRD025-PDE3A) and H572A PDE3A, which does not support DNMDP response[3], each transduced at a multiplicity of infection (MOI) of 1.

**DMS cellular screen.** In all, 92.5 million GB1-6F12 cells were mixed with the PDE3A DMS library virus in E10 media + 8 μg/ml polybrene, resulting in a 40% infection rate and MOI of 1. Infected cells were split to 6-well plates at 1.5 million cells/well and centrifuged at $931 \times g$ (2000 rpm) for 2 h at 30 °C. An aliquot of 1.5 million cells in E10 + 8 μg/ml polybrene served as a control. Five hours after virus addition, media was replaced with fresh E10 + 1× Pen/Strep. On day 2, cells were harvested and split into replicates (2× 6-well plates per replicate) and 10 million cells were plated per T175 flask in 30 ml E10. Cells were treated with 1 μg/ml puromycin for 4 days, after which time $T_0$ cells representing the initial, pretreatment state were harvested and compound treatment was initiated. Treatment arms included DMSO, 10 nM DNMDP, 100 nM DNMDP, and 100 nM trequinsin, the potent but non-cytotoxic PDE3A inhibitor. Following 5 days of compound treatment, surviving cells were harvested and lysed, and genomic DNA (gDNA) was isolated.

**DMS screen deconvolution.** In order to extract PDE3A ORFs from gDNA of surviving cells, 96 PCR reactions were done per gDNA sample. Each PCR reaction was performed with the Q5 DNA polymerase (New England Biolabs) in 50 μl with 800 ng gDNA. One-third of the 96 PCR reactions for each gDNA sample were pooled, concentrated with the Qiagen PCR Cleanup Kit, and purified by running on a 1% agarose gel. The excised bands were purified first by Qiagen Qiaquick kits, followed by the AMPure XP Kit (Beckman Coulter).

Following the Illumina Nextera XT protocol, for each sample we set up 6 Nextera reactions, each with 1 ng of purified ORF DNA. Each reaction was indexed with unique i7/i5 index pairs. After the limited-cycle PCR step, the Nextera reactions were purified with the AMPure XP Kit. All samples were then pooled and sequenced using the Illumina Novaseq S4 platform.

NovaSeq S4 data were processed with the software "AnalyzeSaturationMutagenesis" developed by Broad Institute (Yang et al., manuscript in preparation). Typically, the pair-end reads were aligned to the reference sequence. Multiple filters were applied and some reads were trimmed. The counts of detected variants were tallied. The output files from AnalyzeSaturationMutagenesis, one for each screening sample, were parsed, annotated, and merged into a single.csv file ready for hit-calling.

**DMS data analysis.** For each replicate, the fraction of altered reads was calculated for each individual variant at $T_0$ and at day 5. Variants with $T_0$ count fractions in the bottom 8% were considered to be low confidence and were excluded from downstream analysis. Treatment conditions were then normalized to $T_0$ and averaged across replicates. Average drug treatment versus compound control LFCs were then calculated using DMSO or trequinsin as control treatments. Z scores were calculated for 100 nM DNMDP versus DMSO LFC to determine significance.

**Plasmid generation, cell culture, and viability assays.** Primers used for mutagenesis are listed in Supplementary Table 5. All plasmids were sequence-verified along the entire ORF length for the presence of desired mutation and the absence of additional unwanted mutations.

A pDONR-SLFN12 plasmid from the TRC consortium (ccsbBroadEn_08470) and the GeneArt™ PLUS Site-Directed Mutagenesis System (Thermo Fisher Scientific A14604) were used to generate the C-terminal truncated and the active site mutant SLFN12 ORFs. PDE3A mutant ORFs were generated using the pDONR-PDE3A plasmid (TRC consortium, ccsbBroadEn_06701) and overlapping PCR followed by Gateway BP recombination into pDONR223 (Broad Institute

GPP and BP Clonase II, Thermo Fisher Scientific 11789020). Mutant ORFs were then shuttled into the pLX304 lentiviral expression vector using Gateway LR Clonase II (Thermo Fisher Scientific 11791020) for constitutive expression in cell lines after lentiviral delivery. In addition, SLFN12 active site mutant ORFs were also shuttled into pLX307 and used in co-immunoprecipitation experiments after transient transfection into HeLa cells.

For constitutive expression of 3 × FLAG-tagged SLFN12, a pLX307-3×FLAG expression vector was generated by cloning a pair of annealed oligos containing an in-frame 3×FLAG sequence into the SpeI and EcoRV sites in pLX307, replacing the V5-tag sequence. The SLFN12 ORF sequence was then shuttled from pDONR-SLFN12 (ccsbBroadEn_08470) into pLX307-3xFLAG to allow EF1alpha promoter-driven expression in cells after transfection.

For doxycycline-induced SLFN12 expression, pTetOn-GW-3xFLAG, a Gateway compatible derivative of pTetOne-puro (Takara 634311), was made by inserting an attP-ccdb-Chloramphenicol-resistance cassette plus C-terminal in-frame 3×FLAG tag sequence using Gibson assembly (NEB E2611L). Entry clones of SLFN12 with a proper Kozak consensus sequence were generated based on ccsbBroadEn_08470, first reversing two common single-nucleotide polymorphisms (S43R and C168R) back to S and C, respectively, using the GeneArt™ PLUS Site-Directed Mutagenesis System, and then adding Kozak sequences using standard or overlapping PCR with a Kozak-containing forward primer followed by Gateway BP cloning into pDONR223. SLFN12 PIR truncation mutants were generated using overlapping PCR followed by Gateway BP cloning into pDONR223. Wild-type and mutant SLFN12 ORFs were then shuttled into pTetOn-GW-3xFLAG for inducible expression in cell lines after lentiviral delivery.

**Cell culture and viability assays.** For lentivirus-based gene delivery, HEK293T cells were transfected with ORF overexpression constructs and packaging plasmids psPAX2 and pMD2.G. Virus was collected 48 h after transfection and added to target cells for spin infection with 8 μg/ml of polybrene. Transduced target cells were selected using 15 μg/ml blasticidin (pLX304 series) or 1 μg/ml puromycin (pTetOn-GW-3xFLAG series) and then expanded for 3 days before plating into 384-well plates at 500 cells per well for DNMDP sensitivity testing. DNMDP (Life Chemicals F1638-0042) and trequinsin (Sigma-Aldrich T2057) were added 24 h after plating. For experiments using doxycycline induction, 2 μM doxycycline hydrochloride (Sigma Aldrich D3072) was immediately added after cell plating. After 72 h of DNMDP treatment, cell viability was measured by Cell Titer Glo (Promega G7573).

**Linker resin pulldown.** Linker resin pulldown to measure the binding of DNMDP to wild-type and mutant PDE3A proteins in A2058 PDE3A KO cells were performed as previously described[1,3]. Briefly, cells were lysed in modified RIPA lysis buffer (150 mM NaCl, 10% glycerol, 50 mM Tris-Cl pH 8.0, 50 mM MgCl₂, 1% NP-40) supplemented with EDTA-free protease inhibitors (Sigma-Aldrich 4693159001) and PhosSTOP phosphatase inhibitors (Sigma-Aldrich 4906837001). Two micrograms of total protein (0.5 mg/ml, 400 μl) was incubated with 3 μl DNMDP-2L affinity linker resin[1] with or without 10 μM trequinsin for 4 h at 4 °C. Beads were then washed three times with lysis buffer and bound proteins were eluted with 50 μl LDS sample loading buffer with reducing agent added (Thermo Fisher Scientific NP0007 and NP0009). Twenty microliters of each eluted sample and 50 μg of corresponding matched input lysate (50% input) were separated on 4–12% Bis-Tris PAGE gels (Thermo Fisher Scientific) and immunoblotted with anti-PDE3A antibody (1:1000, Bethyl A302-740) as described above. The same gel was also probed with an anti-Actin antibody (1:4000, Cell Signaling Technology CST 3700) as a loading control.

**PDE3A-SLFN12 complex formation assay.** Complex formation between PDE3A mutant alleles and SLFN12 (Fig. 6d) was assayed in HeLa-PDE3A CRIPSR knockout cells co-transfected with plasmids expressing V5-tagged wild-type or mutant PDE3A and 3 × FLAG-tagged SLFN12 at a DNA ratio of 1:4. At 72 h post transfection, cells were treated with 10 μM DNMDP or DMSO for 8 h, collected, and snap frozen until lysis. Cell pellets were lysed using modified RIPA buffer as described above. Three micrograms of total protein was immunoprecipitated with anti-V5 magnetic beads (MBL Life Science M167-11) at 4 °C overnight, and the co-precipitated SLFN12 protein was assayed by anti-FLAG immunoblot (1:2000, Sigma-Aldrich M2 antibody F1804) and anti-V5 immunoblot (1:5000, Thermo Fisher Scientific R960-25).

To assay complex formation between PDE3A and wild-type or active site mutant SLFN12 (Fig. 7a), HeLa cells were transfected with ORF overexpression constructs expressing V5-tagged SLFN12 (TRC consortium, TRCN0000476272) or pLX307-SLFN12 mutants using the Fugene 6 transfection reagent (Promega E2691). DNMDP treatment, cell collection, and cell lysis were performed as above. Three micrograms of total protein was incubated with 4 μg of anti-PDE3A antibody (Bethyl 302–741A) at 4 °C overnight, followed by addition of 20 μl each of Protein A- and Protein G-Dynabeads (Thermo Fisher Scientific 10001D and 10003D) and incubation at 4 °C for 2 h. Beads were washed and bound proteins were eluted, subject to SDS-PAGE separation, and anti-PDE3A and anti-V5 immunoblotting as described above.

**Generation and characterization of DNMDP-resistant cell lines.** HeLa cells were cultured in Dulbecco's Modified Eagle Medium + 10% FBS media, grown to confluence in a 10-cm dish, and treated with 1 μM of DNMDP for 2 weeks. Media and DNMDP were refreshed three times a week. Plates were kept in culture until resistant cells proliferated to a number suitable for biochemical assays. Next-generation RNA sequencing revealed complete loss of SLFN12 sequencing reads, and loss of SLFN12 expression was verified by real-time reverse transcription-PCR using TaqMan probes specific to SLFN12.

**RNA cleavage assay.** Total RNA was extracted from HeLa cells using Trizol reagent (Invitrogen) following the manufacturer's protocol. After DNase (Thermo fisher, AM2239) treatment, RNA was cleaned up on RNA Clean & Concentrator-25 (Zymo Research). To induce or inhibit the complex formation, 2.5 μM PDE3A and SLFN12 recombinant proteins were incubated with DMSO (vehicle control), 12.5 μM DNMDP, 12.5 μM trequinsin, or 12.5 μM estradiol at room temperature for 30 min prior to RNA cleavage assay. In the cleavage assay, 2 μg of total RNA was incubated with the indicated concentrations of recombinant proteins in the RNA cleavage buffer containing 40 mM Tris-HCl, pH 8.0, 20 mM KCl, 4 mM $MgCl_2$, and 2 mM DTT at 37 °C for 40 min. RNA samples were analyzed on a denaturing formaldehyde agarose (1%) gel. After electrophoresis, RNA was stained with ethidium bromide and visualized under UV light.

**Chemical crosslinking of PDE3A and SLFN12 proteins using EDC/NHS.** The indicated concentrations of recombinant proteins were incubated at room temperature for 1 h in the RNA cleavage buffer containing 20 mM Hepes, pH 7.4, 10 mM EDC (1-ethyl-3-(3-dimethylaminopropyl)carbodiimide) and 5 mM NHS (N-hydroxysuccinimide) to crosslink adjacent proteins. Crosslinking reaction was stopped by adding SDS-loading buffer. Protein samples were analyzed on 6% SDS-PAGE. After electrophoresis, proteins were silver-stained and visualized using the SilverQuest Silver Staining Kit (ThermoFisher) following the manufacturer's instructions.

**Statistics and reproducibility.** The results from the MBP-SLFN12 pulldown experiments with PDE3A$^{CAT}$ shown in Fig. 1d were independently confirmed in at least two additional experiments. For Fig. 5b, the ability of SLFN12 C-terminal truncation mutants to mediate DNMDP sensitivity was independently tested in A549 + pPDE3A cells, which ectopically express PDE3A. Identical results were observed. For Fig. 6c, the linker resin pulldown experiment was independently carried out in HeLa PDE3A CRISPR KO cells and consistent results were observed. For Fig. 6d, complex formation was independently tested using the SLFN12 catalytic mutant, E200A/E205A, with identical results observed. For Fig. 7a, complex formation between PDE3A and SLFN12 active site mutants was independently assayed using doxycycline-induced SLFN12 expression in HeLa-Res cells lacking endogenous SLFN12 expression. Consistent results were observed. For Fig. 7c–e, in vitro RNA cleavage assays were independently performed three times each with the indicated concentration of recombination proteins and compounds. Consistent results were observed.

**Reporting summary.** Further information on research design is available in the Nature Research Reporting Summary linked to this article.

## Data availability

PDB accession numbers for the crystal structures in Fig. 2 are: PDE3A + cAMP, 7L29 [https://doi.org/10.2210/pdb7L29/pdb]; PDE3A + DNMDP, 7KWE [https://doi.org/10.2210/pdb7KWE/pdb]; PDE3A + trequinsin, 7L28 [https://doi.org/10.2210/pdb7L28/pdb]; PDE3A apo structure, 7L27 [https://doi.org/10.2210/pdb7L27/pdb]. The HDX summary data for Fig. 2 can be found in Supplementary Data 1. The original mass spectra for Fig. 2 and the protein sequence database used for searches have been deposited in the public proteomics repository MassIVE (http://massive.ucsd.edu) and are accessible at ftp://massive.ucsd.edu/MSV000087620/. The PDB accession numbers for the Cryo-EM structures in Figs. 3 and 4 are: PDE3A, 7LRC [https://doi.org/10.2210/pdb7LRC/pdb]; SLFN12, 7LRE [https://doi.org/10.2210/pdb7LRE/pdb]; PDE3A-DNMDP-SLFN12 complex, 7LRD [https://doi.org/10.2210/pdb7LRD/pdb]. The EMDB accession numbers for the Cryo-EM structures in Figs. 3 and 4 are: PDE3A, EMD-23494; SLFN12, EMD-23496; PDE3A-DNMDP-SLFN12 complex, EMD-23495. The raw data for the deep mutational scanning experiment in Fig. 6 can be found in Supplementary Data 2. Source data are provided with this paper.

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

## Acknowledgements

We thank Dr. Alex Burgin, Dr. Nicole Persky, and Dr. Elisa Aquilanti for valuable discussions. The coordinates for the four crystal structures have been deposited in the PDB database with accession codes 7L27, 7L29, 7KWE, and 7L28. See data availability section for accession numbers for Cryo-EM structures. This work was supported by funding from Bayer Pharmaceuticals, the Austrian Marshall Plan Foundation (to M.T.), and a Broad Institute NextGen award (to H.G.).

## Author contributions

C.W.G., X.W., M.P., S.L., J.F., G.R.S., S.W.H., A.B., T.Z., J.P.M., M.T., M.J.R., L.d.W., J.M., B.K., D.R., and A.T. conducted the experiments. C.W.G., X.W., L.W., S.H.H., A.D.C., and H.G. analyzed the DMS data. C.W.G., X.W., M.P., S.L., J.F., G.S., F.P., X.Y., C.T.L., and H.G. designed the experiments. C.W.G., X.W., M.P., S.L., G.R.S., M.L., T.A.L., S.A.C., A.D.C., C.T.L., M.M., and H.G. developed the scientific strategy. C.W.G., X.W., M.P., S.L., C.T.L., M.M., and H.G. wrote the paper.

## Competing interests

Several authors received funding from Bayer AG: C.W.G., X.W., S.L., G.S., S.W.H., A.B., L.W., S.H.H., J.M., B.K., T.L., C.L., M.M., H.G. A.T. is currently an employee of Bayer AG. M.L. is a former employee of Bayer AG and current employee of Nuvisan ICB GmbH. F.P. is currently employed by Merck Research Laboratories. M.M. also receives research funding from Janssen, Novo, and Ono; consulting fees from Bayer, Interline, and OrigiMed; and royalties from LabCorp. In addition, X.W., L.d.W., T.A.L., M.M., and H.G. receive an inventors' share of license revenue as part of their employment for certain patent filings, including US-2016-0016913 and US-2018-0235961, which relate to aspects of the work described in this manuscript. The co-owners of those patent filings are The Broad Institute, Inc., Dana-Farber Cancer Institute, Inc., and Bayer Pharma AG. The other authors declare no competing interests.
