## [Peer Review File · Nature Communications]

REVIEWER COMMENTS

Reviewer #1 (Remarks to the Author):

In their earlier studies, the authors reported that the compound DNMDP binds to phosphodiesterase 3A (PDE3A), and promotes the interaction between PDE3A and the Schlafen family member SLFN12, thereby exerting a tumor cell killing effect. In this manuscript, the authors pushed this story forward by presenting crystal structures of the catalytic domain of PDE3A (PDE3A-CAT) with several compounds, and cryo-EM structures of PDE3A-CAT/SLFN12 complex in the presence of DNMDP. The complex structure, as a major conceptual advance of this study, revealed how DNMDP promotes the physical association between PDE3A-CAT and SLFN12. The authors then properly analyzed and verified the important residues at various interfaces of the complex found in the structure by biochemical and cell biology assays. The authors showed that the RNase activity of SLFN12 is In addition, this is the first structural report for the SWADL domain of the Schlafen family. Overall, this is a well-conducted study valuable for the development of new cancer therapies.

Comments:

1. Figure 1e. Several chromatographs are pretty much the same as those in Figure 1a. Addition of DNMDP actually had basically no effect on complex formation between PDE3A and SLFN12. I don't see much information carried by this figure. It may be better to put Figure 1e in the supplementary information.

2. In their previous work, the authors mentioned that aryl hydrocarbon receptor-interacting protein (AIP) is required for PDE3A-SLFN12 complex formation. They found that "AIP knockout completely abolished PDE3A-SLFN12 complex formation in response to DNMDP" by IP experiments in HeLa cells. Yet in this manuscript the PDE3A-SLFN12-DNMDP complex seemed quite stable in solution. Any explanation?

3. Line 145~147, "In contrast, no change of T_m ...". Although this is not wrong in 99% of the cases, a negative result from ITC or BLI assays would be better evidence to show "SLFN12 could not bind DNMDP".

4. Figure 3c. The structural presentation of the dimeric interface of SLFN12 is unclear. The residues shown in the stick model are too small and the labels with prime marks are somewhat misleading. It also looks a bit weird to remove most part of a monomer and leave only some secondary structural elements. Consider moving some panels from Supp Figure 5 to Figure 3c.

5. Figure 4d. Better indicate the rotation operations as in Figures 2a and 3a.

6. The authors spent quite some effort to screen potential DNMDP resistance mutations on the catalytic domain of PDE3A using an artificial library of PDE3A alleles. It would be interesting to know whether mutations on the corresponding region of PDE3A are implicated in cancer patients. Perhaps the authors can do a TCGA search for that.

7. Any structure-based explanation for how complexing with PDE3A and DNMDP stimulates the RNase activity of SLFN12?

Reviewer #2 (Remarks to the Author):

Dear authors,

with great interest, I read your manuscript entitled "Structure of PDE3A-SLFN12 complex reveals requirements for activation of SLFN12 RNase", submitted to Nature Communications. The manuscript reports on the importance of joint action of the phosphodiesterase 3A (PDE3A), the SFLN12 protein, and the novel PDE3A inhibitor, the DNMDP molecule, for induced death of cancer cell. In particular, authors provide high-resolution x-ray and cryo-EM models of the catalytic domain of PDE3A and of the SLFN12 protein with/without presence of inhibitors or AMP, which enable understanding of the joint action of the three entities at the structural level. The structural analysis is backed up and extended by a broad range of solid biochemical and biophysical experiments. Altogether, the authors propose that DNMDP promotes requisition of two SLFN molecules by optimizing their mutual steric orientations, promoting thus their RNAase activity and thereby killing cancer cells.

I believe that the research is very solid. Still, I have a few concerns:

1) Catalytic domain of PDE3 vs full-length protein:

Most of your analyses is performed with isolated C-terminal part of PDE3 (PDE3cat), while the deep mutational analysis is performed with full-length PDE3. This suggests that the results acquired with PDE3cat could be extended to the full PDE3 molecule, but the effect of including of the preceding +600 AA isn't discussed in your manuscript, even if they form more than a half of the full PDE3A. I believe the paper would greatly profit if you could extend it this way, e.g. discussing these questions:

- a) What was the motivation to work with the selected AA range of PDE3A?
 - b) What is the structure and function of the first +600 AA of PDE3A, and how could it influence your measurements/conclusions? References to earlier results would be nice here.
- Otherwise, I suggest to limit your conclusions to the "catalytic domain of PDE3A".

2) Unmodeled loops of PDE3A, omitted in crystal structures:

- a) Please, indicate why the two loops were replaced as stated (only diffraction quality?)
- b) The loops should be at least partially detectable in the cryoEM map. It is possible, though, that they are unclear due to the C2 symmetry imposed. It may be worth to run another 3D refinement in C1 symmetry, with particles included in the last refinement prior to multi-body refinement.
- c) Their location with respect to the SLFN12 molecules is unclear - Fig. 2a focuses on PDE3A model and 3a on SLFN12. Perhaps it would suffice to show a one PDE3A surface model in Fig. 3a in cartoon as in Fig. 2a. Could they possibly influence the SLFN12 molecules in the vicinity of the detected RNAase active sites E200/E205? Are there any unmodeled densities?

3) The structural effect of DNMDP binding to the hydrophobic pocket in the PDE3A catalytic domain seems to be very subtle but enormously important and stable even at high salt conditions, which seems to be its advantage.

- a) Interestingly, the four PDE3xtal crystal structures show little difference upon binding of different compounds. But what is the difference between the cryoEM model and the crystal structures, the "apo" in particular? It would deserve addition to Supp. Fig. 1.
- b) The salt conditions should be indicated in the main text in the RNAase activity paragraph (p. 13, l. 331-).

4) Fig. 4d: The position of the mutated residues with respect to PDE3 is unclear. Again, a composite ribbon+surface figure of both proteins would be more informative.

5) Fig. 5C, 6c,d: please provide more detailed description of gel bands. 6a: include LFC meaning in legend.

Minor points:

p. 4, l.83: indicate the range of AA

p. 6, l. 152: indicate AA range of replaced loops

p. 9, l. 230: include the AA range of replaced loops in crystal structure

p. 11, l. 270: ...sidechains. -> sidechains of SLFN12.

Conclusions, p. 14, l. 355-359. On l. 355, "act to induce dimerization" should be changed to "promote" only, since both PDE3Acat and SLFN12 happily bind already at 150 mM NaCl (Fig. 1e), and the complex without DNMDP can cleave RNA already at only slightly different salt conditions (4mM MgCl₂+20 mM KCl, Fig. 7d).

Methods, cryo-EM data processing: the meaning of "couple to" is unclear without inspecting Supp. fig. 3 - when and how many 3D refinements were performed. Please rephrase.

I am convinced that your research is interesting for a broad range of readers of Nature Communications.

With best regards!

Reviewer #3 (Remarks to the Author):

I really like this paper in terms of the biophysical craft and the combination of different techniques from equilibrium hydrodynamic measurements to extract dissociation constants, cryo-EM for structural insight and HDX-MS for dynamics. The combination of these different methods has put the authors into a very strong position to support their arguments. The use of HDX-MS is appropriate, and the authors have gone to great lengths to solve the protein structures prior to the application of HDX and are therefore in a good position to interpret the data. The manuscript contains important and frequently overlooked details such as the fraction bound in the main text which has important ramifications on the interpretation of HDX-MS difference data.

The absence of extraneous exchange controls is appropriate for this kind of qualitative work. However, comments such as "A dynamic exchange behavior, evident by extensive deuteration at the earliest time point, was observed in both the... .. which indicates they are unstructured" is a bit casual and cannot be reasonably made. This is because it is impossible to quantify the extent of exchange unless 1) the data have been corrected for extraneous exchange 2) some form of modelling on the corrected data has been performed to extract information from the peptides that is consistent with protein unstructure. The authors should refrain from interpretations of their HDX-MS data that require quantification of the HDX rate constants.

I could not find details of how the authors converted the original peptide x-axis data to amino acid x-axis data. I suppose some form of averaging will have been required in order to map the HDX-MS data onto the structures and details should be provided about how this was done. There is also no interpretation or mention of allosteric effects that can result in changes in HDX-MS profile that are distal to any binding site and are therefore only indirectly associated with binding. At the very least the authors should mention this in the text and go some way to discount it if it doesn't agree with their original interpretation.

We thank the reviewers for their insightful comments and hope that the following responses satisfy their concerns!

Reviewer 1

1. Figure 1e. Several chromatographs are pretty much the same as those in Figure 1a. Addition of DNMDP actually had basically no effect on complex formation between PDE3A and SLFN12. I don't see much information carried by this figure. It may be better to put Figure 1e in the supplementary information.

We agree with the reviewer that the figures do appear at first glance to be redundant, although they do contain different information. Figures 1E and 1F are necessary in the main figure in order to do a direct comparison of the effect of [NaCl] on complex formation. Figure 1A contains the experimental information related to determining the mass of the protein and protein complexes in solution. We do discuss the solution mass in the text so in order to remove the redundancy between 1A and 1E, but preserve the comparison of 1E and 1F, Figure 1A has been moved to the supplementary and the figure numbers updated appropriately.

2. In their previous work, the authors mentioned that aryl hydrocarbon receptor-interacting protein (AIP) is required for PDE3A-SLFN12 complex formation. They found that "AIP knockout completely abolished PDE3A-SLFN12 complex formation in response to DNMDP" by IP experiments in HeLa cells. Yet in this manuscript the PDE3A-SLFN12-DNMDP complex seemed quite stable in solution. Any explanation?

AIP is required for complex formation in cells, but not in vitro. As shown in our previous paper (Wu et al., 2020), the requirement of AIP for DNMDP sensitivity can be overcome by ectopically expressing high levels of PDE3A, implying that the primary function of AIP is stabilization of endogenous PDE3A. This function is evidently not necessary when working with purified proteins in vitro. In response to this comment, we have added to the last paragraph of the first Results section, "*The co-chaperone, Aryl Hydrocarbon Receptor Interacting Protein (AIP), is required for PDE3A-SLFN12 complex formation in cells. However, AIP is evidently not required for complex formation in vitro.*"

3. Line 145~147, "In contrast, no change of T_m...". Although this is not wrong in 99% of the cases, a negative result from ITC or BLI assays would be better evidence to show "SLFN12 could not bind DNMDP".

The reviewer brings up a good point and having an orthogonal method such as BLI and/or ITC to support our observation would be ideal. Studying protein-compound interactions with BLI is very challenging due to the small size of the compound relative to the immobilized protein, making it difficult to detect. In our experience we don't believe it serves as a good orthogonal method to study compound binding. We do routinely use ITC and we did attempt to study complex formation with this approach. While we could get clear binding data for the compound interacting with PDE3A, we could never achieve a stable baseline when SLFN12 was in the cell

and titrated with buffer, compound, or PDE3A. We attributed this to the poor behavior of isolated SLFN12 at higher concentrations required for ITC. Hence, we focused on BLI to study the protein-protein interactions. To address the reviewer's comments, we have modified the text on page 6 to state the following:

"In contrast, no change in T_m was observed upon incubating SLFN12 with DNMDP (Supplementary Fig. 1). This suggests that SLFN12 does not bind to DNMDP in the absence of PDE3A, although we cannot discount the possibility that its binding does not sufficiently impact the stability of the structure to cause a change in the melting temperature of SLFN12."

4. Figure 3c. The structural presentation of the dimeric interface of SLFN12 is unclear. The residues shown in the stick model are too small and the labels with prime marks are somewhat misleading. It also looks a bit weird to remove most part of a monomer and leave only some secondary structural elements. Consider moving some panels from Supp Figure 5 to Figure 3c.

We appreciate the feedback on the figures. We removed the current Figure 3C and replaced it with the more detailed figures from Figure S5E.

5. Figure 4d. Better indicate the rotation operations as in Figures 2a and 3a.

The figures have been annotated to reflect the rotation operations.

6. The authors spent quite some effort to screen potential DNMDP resistance mutations on the catalytic domain of PDE3A using an artificial library of PDE3A alleles. It would be interesting to know whether mutations on the corresponding region of PDE3A are implicated in cancer patients. Perhaps the authors can do a TCGA search for that.

We thank the reviewer for this forward-thinking suggestion. We have cross-referenced TCGA pan-cancer mutation analysis and identified PDE3A mutations that scored significantly in our deep mutational scanning experiment. PDE3A is mutated at low levels in TCGA (356 mutation incidents detected in 313 tumor samples, mostly not recurrent). Of these, a small number (21 mutations in 23 tumor samples) scored significantly in the DMS screen. These results have been added to Supplementary Dataset 2.

7. Any structure-based explanation for how complexing with PDE3A and DNMDP stimulates the RNase activity of SLFN12?

As mentioned in the Conclusion section, we propose that the key function of velcrins is to induce SLFN12 dimerization, perhaps analogously to activation of RNaseL by 2'-5'-linked oligoadenylates (Han et al., Science, 2014). However, we are not yet sure how this results in stimulation of SLFN12 RNase activity. We are currently working on addressing this question, but it is beyond the scope of the current manuscript. To address this comment, we have added into the Conclusion, *"We hypothesize that dimerization of SLFN12 may stimulate SLFN12 RNase activity, analogously to activation of RNaseL by 2'-5'-linked oligoadenylates."*

Reviewer 2

1) Catalytic domain of PDE3 vs full-length protein:

Most of your analyses is performed with isolated C-terminal part of PDE3 (PDE3cat), while the deep mutational analysis is performed with full-length PDE3.

This suggests that the results acquired with PDE3cat could be extended to the full PDE3 molecule, but the effect of including of the preceding +600 AA isn't discussed in your manuscript, even if they form more than a half of the full PDE3A. I believe the paper would greatly profit if you could extend it this way, e.g. discussing these questions:

a) What was the motivation to work with the selected AA range of PDE3A?

We selected the PDE3A catalytic domain for deep mutational scanning because we showed in earlier work (Wu et al., JBC, 2020) that the isolated catalytic domain was sufficient to confer DNMDP sensitivity in cells lacking endogenous PDE3A but expressing SLFN12. We agree that this was not clear in the text and have added the following sentence to this section: *"Because we previously showed that the isolated catalytic domain of PDE3A was sufficient to confer DNMDP sensitivity in cells expressing SLFN12 but lacking endogenous PDE3A (Wu et al), we limited our mutational analysis to the PDE3A catalytic domain."*

b) What is the structure and function of the first +600 AA of PDE3A, and how could it influence your measurements/conclusions? References to earlier results would be nice here. Otherwise, I suggest to limit your conclusions to the "catalytic domain of PDE3A".

The first 600 amino acids of PDE3A contain several membrane-associated domains (Kenan et al., JBC, 2000). Because we have previously shown that the isolated catalytic domain of PDE3A functions similarly to full-length PDE3A in supporting DNMDP sensitivity (Wu et al., JBC, 2020), we reasoned that we could remove the hydrophobic N-terminus from our in vitro expression constructs without affecting complex formation. Interestingly, short forms of PDE3A without the membrane-associated domains can also be expressed from alternative start sites in the full-length cDNA, leading to the multiple bands seen by immunoblotting for endogenous PDE3A or ectopically-expressed full-length PDE3A constructs (Wechsler, J Biol Chem, vol 277, p 38072, 2002; and Vandeput, Proc Natl Acad Sci, vol 110, p 19778, 2013).

We agree with the reviewer that this was not clear in the text, and we have made the following modifications in the first paragraph of the Results section for clarification:

"We limited our analysis to the catalytic domain of PDE3A because our previous experiments indicated that the N-terminal portion of PDE3A, containing several membrane association domains (Kenan et al), was not required for DNMDP sensitivity in cells (Wu et al)."

2) Unmodeled loops of PDE3A, omitted in crystal structures:

a) Please, indicate why the two loops were replaced as stated (only diffraction quality?)

The text on page 6 has been modified to include the following:

"PDE3A^{CAT-Xtl} is comprised of residues 669 to 1095 with two internal loops between residues 780-800 and 1029-1067 replaced with shorter linkers to aid in crystallization and improve diffraction quality of the crystals (Fig. 2a)."

b) The loops should be at least partially detectable in the cryoEM map. It is possible, though, that they are unclear due to the C2 symmetry imposed. It may be worth to run another 3D refinement in C1 symmetry, with particles included in the last refinement prior to multi-body refinement.

No density was evident in the cryo-EM maps for these loop regions. 3D refinement in C1 was performed at different stages during the refinement (please see supplemental Figure 3S) but no improvement in density was observed that would allow further building of these loop regions. The reviewer does bring up a valid point in that we do not address these loops and their potential role in complex formation. We have addressed this in reply to comment "c)" following.

c) Their location with respect to the SLFN12 molecules is unclear - Fig. 2a focuses on PDE3A model and 3a on SLFN12. Perhaps it would suffice to show a one PDE3A surface model in Fig. 3a in cartoon as in Fig. 2a. Could they possibly influence the SLFN12 molecules in the vicinity of the detected RNase active sites E200/E205? Are there any unmodeled densities?

The reviewer quite correctly points out we did not address the likely location and any potential role of the loop regions in the cryo-EM structure. To address their location in Figure 3A dashed lines have been indicated in the left most PDE3A monomer to indicate their location relative to SLFN12 along with the last residue visible in the electron density. We attempted a number of variants on surface/cartoon representations but this remained the clearest representation of the complex. The figure legend for 3A has been modified to include the following information:

"The loop regions of PDE3A between 779-799 and 1029-1068, which do not show density in the cryo-EM maps, are indicated by dashed lines on one PDE3A monomer."

To address the location of these loops relative to SLFN12 and their potential role in complex formation, the following has been added to Page 10:

" There was no density evident for the loop regions between residues 779-799 and 1029-1068, suggesting that they adopt multiple conformations and are not involved in contacting SLFN12. In the HDX-MS studies these regions showed a high deuterium content at the earliest time point in the absence of SLFN12, indicating that they are dynamic (Fig. S2a). The deuterium uptake did not change in the presence of SLFN12, supporting the observation that they are not affected by complex formation (Fig. S2b and Fig S2c). "

3) The structural effect of DNMDP binding to the hydrophobic pocket in the PDE3A catalytic domain seems to be very subtle but enormously important and stable even at high salt conditions, which seems to be its advantage.

a) Interestingly, the four PDE3xtal crystal structures show little difference upon binding of different compounds. But what is the difference between the cryoEM model and the crystal structures, the "apo" in particular? It would deserve addition to Supp. Fig. 1.

In the manuscript we state that "The structure of PDE3A^{CAT} is essentially the same as PDE3A^{CAT-xtl} (RMSD for backbone atoms of 0.38 Å), with only an additional two turns of the C-terminal helices for each monomer modeled.". We do readily agree with the reviewer that including a comparison of the cryo-EM and the crystal structures will help support these observations and an additional figure has been added (Supplementary Figure S5a).

b) The salt conditions should be indicated in the main text in the RNAase activity paragraph (p. 13, l. 331-).

We used the same salt conditions to measure SLFN12 RNase activity as were used to measure the RNase activity of SLFN13 in a previous study (Yang, J.Y. et al, Nat. Comm., 2018). As requested, we now include the salt conditions in the main text.

"To determine whether SLFN12 is indeed an RNase, we incubated 2 μM recombinant SLFN12 with human rRNA isolated from HeLa cells in a buffer containing 40 mM Tris-HCl, pH 8.0, 20 mM KCl, 4 mM MgCl₂ and 2 mM DTT."

4) Fig. 4d: The position of the mutated residues with respect to PDE3 is unclear. Again, a composite ribbon+surface figure of both proteins would be more informative.

As the reviewer points out, indicating the location of the two catalytic residues is important. To address the reviewer's concern we have included a cartoon representation showing the location of these residues in the supplementary section (Supplementary Figure S5d).

5) Fig. 5C, 6c,d: please provide more detailed description of gel bands. 6a: include LFC meaning in legend.

Fig. 5c and 6c,d have been updated with arrows pointing to the PDE3A or SLFN12 gel bands. Fig. 6a legend has been updated to include the meaning of LFC as "average log₂ fold change".

Minor points:

p. 4, l.83: indicate the range of AA

The following text has been added: "... we expressed and purified the catalytic domain of PDE3A (PDE3A^{CAT}), which comprises residues 640 to 1141,..."

p. 6, l. 152: indicate AA range of replaced loops

The text has been modified to the following: "*PDE3A^{CAT-Xtl} is comprised of residues 669 to 1095 with two internal loops between residues 780-800 and 1029-1067 replaced with shorter...*"

p. 9, l. 230: include the AA range of replaced loops in crystal structure

We have added the following sentence to this section:

"These loop regions are equivalent to the ones that were replaced by short linkers in the crystal structure."

p. 11, l. 270: ...sidechains. -> sidechains of SLFN12.

The text has been modified to say: "*....sterically clash with the L554, I557, and I558 sidechains of SLFN12, making the*"

Conclusions, p. 14, l. 355-359. On l. 355, "act to induce dimerization" should be changed to "promote" only, since both PDE3Acat and SLFN12 happily bind already at 150 mM NaCl (Fig. 1e), and the complex without DNMDP can cleave RNA already at only slightly different salt conditions (4mM MgCl₂+20 mM KCl, Fig. 7d).

We agree with this point and have changed the wording in the Conclusion section as specified.

Methods, cryo-EM data processing: the meaning of "couple to" is unclear without inspecting Supp. fig. 3 - when and how many 3D refinements were performed. Please rephrase.

This sentence has been simplified to the following: "*The resulting 343,715 particles underwent two cycles of 3D refinement and CTF parameter refinement.*"

Reviewer 3

The absence of extraneous exchange controls is appropriate for this kind of qualitative work. However, comments such as "A dynamic exchange behavior, evident by extensive deuteration at the earliest time point, was observed in both the... .. which indicates they are unstructured" is a bit casual and cannot be reasonably made. This is because it is impossible to quantify the extent of exchange unless 1) the data have been corrected for extraneous exchange 2) some form of modelling on the corrected data has been performed to extract information from the peptides that is consistent with protein unstructure. The authors should refrain from interpretations of their HDX-MS data that require quantification of the HDX rate constants.

We fully agree with the reviewer that proper full deuteration controls would be required to correct the HDX data and enable the measurement of HDX rate constants and subsequent

protection factors. The purpose of our HDX-MS experiment was to detect changes of PDE3A deuterium levels upon binding to SLFN12 and as the reviewer correctly points, our HDX-MS data are qualitative. To address the reviewer's point, we edited the main text as follows:

"Extensive deuteration at the earliest time point was observed in both the N- and C-terminals of the protein and in the loop regions between 778-793 and 1028-1068, which indicates dynamic regions."

I could not find details of how the authors converted the original peptide x-axis data to amino acid x-axis data. I suppose some form of averaging will have been required in order to map the HDX-MS data onto the structures and details should be provided about how this was done.

We thank the reviewer for bringing this up, as details of this process are not given in the manuscript. To generate the heat map shown onto Supplementary Figure 2b from the individual peptides identified, HDExaminer first divides the protein into non-overlapping "atomic peptides". These atomic peptides are formed by dividing the protein everywhere an observed peptide starts or ends. Each observed peptide's deuteration level can then be expressed as a sum of deuteration levels for one or more atomic peptides. It should be noted that the first two residues of each peptide are ignored, since these are widely considered to exchange too rapidly. HDExaminer then computes a deuteration level for each atomic peptide that minimizes the least squares error with the set of observed peptides. The least squares calculation was further altered slightly to ensure that calculated deuteration levels did not vary wildly from one residue to the next, a feature of the software known as "smoothing". The heat map algorithm is similar to the one described in *Smith DM & Babić D, 2017, "Localization improvement of deuterium uptake in hydrogen/deuterium exchange in proteins", Journal of Chemometrics, vol 31(3), e2876*

We added the following text to the legend of Supplemental Figure 2.

"The heat map was generated in HDExaminer as follows: First, the protein was divided into non-overlapping "atomic peptides". These atomic peptides are formed by dividing the protein everywhere an observed peptide starts or ends. Each observed peptide's deuteration level was then expressed as a sum of the deuteration levels for one or more atomic peptides. The first two residues of each peptide were ignored, since these are widely considered to exchange too rapidly. The deuteration level for each atomic peptide was subsequently computed by minimizing the least squares error within the set of observed peptides. For mapping onto the heat map and the PDE3A structure, deuteration levels were further smoothed in HDExaminer."

There is also no interpretation or mention of allosteric effects that can result in changes in HDX-MS profile that are distal to any binding site and are therefore only indirectly associated with binding. At the very least the authors should mention this in the text and go some way to discount it if it doesn't agree with their original interpretation.

At the experimental conditions employed in this study (deuterium labeling at 20 °C), we did not detect any long-distance allosteric changes on PDE3A upon binding of SLFN12, however, we cannot rule out the possibility that these may occur. Further, deuterium changes detected in the PDE3A dimer interface may be considered allosteric, given that these are distal from the SLFN12 binding site. We modified the text as follows to include a comment about the detection of allostery in our experiments.

“This suggests that the binding of SLFN12 stabilizes the PDE3A^{CAT} homodimer further, presumably via the reduction of structural fluctuations in the dimer interface, not captured in the crystal structure. At the experimental conditions employed, we did not detect any long-distance allosteric changes in PDE3A upon interaction with SLFN12 presumably in regions 2 and 3, although the deuterium changes detected in the PDE3A dimer interface may be considered such.”

We also note that we updated Fig. 5 by streamlining our analysis to just the two truncation mutants relevant for investigating PIR function.

REVIEWER COMMENTS

Reviewer #1 (Remarks to the Author):

The authors have adequately addressed my concerns. The manuscript, including the texts and figures, have been substantially improved. I am happy with the currently version for publication in Nature Communications.